# Selective CO₂ electrolysis to CO using isolated antimony alloyed copper

Jiawei Li[1,2,7], Hongliang Zeng[1,7], Xue Dong[3,7], Yimin Ding[4], Sunpei Hu[2], Runhao Zhang[1,2], Yizhou Dai[1,2], Peixin Cui [5], Zhou Xiao[2], Donghao Zhao[2], Liujiang Zhou [4,6], Tingting Zheng[1], Jianping Xiao [3] ✉, Jie Zeng [2] ✉ & Chuan Xia [1,4] ✉

Renewable electricity-powered CO evolution from CO₂ emissions is a promising first step in the sustainable production of commodity chemicals, but performing electrochemical CO₂ reduction economically at scale is challenging since only noble metals, for example, gold and silver, have shown high performance for CO₂-to-CO. Cu is a potential catalyst to achieve CO₂ reduction to CO at the industrial scale, but the C-C coupling process on Cu significantly depletes CO* intermediates, thus limiting the CO evolution rate and producing many hydrocarbon and oxygenate mixtures. Herein, we tune the CO selectivity of Cu by alloying a second metal Sb into Cu, and report an antimony-copper single-atom alloy catalyst (Sb₁Cu) of isolated Sb-Cu interfaces that catalyzes the efficient conversion of CO₂-to-CO with a Faradaic efficiency over 95%. The partial current density reaches 452 mA cm⁻² with approximately 91% CO Faradaic efficiency, and negligible C₂₊ products are observed. In situ spectroscopic measurements and theoretical simulations reason that the atomic Sb-Cu interface in Cu promotes CO₂ adsorption/activation and weakens the binding strength of CO*, which ends up with enhanced CO selectivity and production rates.

The massive emission of CO₂ caused by fossil fuel combustion has led to a dramatic increase in CO₂ concentrations in the atmosphere, which has triggered global concerns about climate change[1-4]. The electrochemical CO₂ reduction reaction (CO₂RR) offers a sustainable approach to directly convert CO₂ into value-added chemicals and fuels under ambient conditions, which reduces CO₂ emissions and alleviates the dependence on fossil fuels[5-9]. Among the products obtained from CO₂RR, CO is one of the most important feedstocks that can be used in the sustainable production of commodity chemicals[10-12]. To perform this reaction economically at scale, a catalyst capable of mediating the efficient formation of CO with high selectivity at high current densities is a prerequisite. Currently, gold and silver are the most active catalysts for this process, with near-unity CO selectivity under low to modest production rates. To maintain high CO selectivity, the partial current

[1]School of Materials and Energy, University of Electronic Science and Technology of China, Chengdu 611731, PR China. [2]Hefei National Research Center for Physical Sciences at the Microscale, Key Laboratory of Strongly Coupled Quantum Matter Physics of Chinese Academy of Sciences, National Synchrotron Radiation Laboratory, Key Laboratory of Surface and Interface Chemistry and Energy Catalysis of Anhui Higher Education Institutes, Department of Chemical Physics, University of Science and Technology of China, Hefei, Anhui 230026, PR China. [3]State Key Laboratory of Catalysis, Dalian Institute of Chemical Physics, Chinese Academy of Sciences, University of Chinese Academy of Sciences, Dalian National Laboratory for Clean Energy, Dalian 116023, PR China. [4]Yangtze Delta Region Institute (Huzhou), University of Electronic Science and Technology of China, Huzhou, Zhejiang 313001, PR China. [5]Key Laboratory of Soil Environment and Pollution Remediation, Institute of Soil Science, Chinese Academy of Sciences, Nanjing, PR China. [6]School of Physics, University of Electronic Science and Technology of China, Chengdu 610054, PR China. [7]These authors contributed equally: Jiawei Li, Hongliang Zeng, Xue Dong. ✉e-mail: xiao@dicp.ac.cn; zengj@ustc.edu.cn; chuan.xia@uestc.edu.cn

densities of CO ($j_{CO}$) of these noble metals are typically lower than 200 mA cm$^{-2}$ [13-15]. Further increasing the bias will simultaneously promote the competitive hydrogen evolution reaction (HER) and thus suppress the CO Faradaic efficiency (FE). Other more abundant, less expensive metals generally have poor selectivity for $CO_2$-to-CO[16,17]. Very recently, earth-abundant molecular electrocatalysts[18,19] and single-atom catalysts[20-22] have been demonstrated to exhibit an FE of CO formation comparable with those of noble metal catalysts. However, most of these catalysts, in which the catalytic metal sites are isolated and well-defined, have also failed to drive $CO_2$-to-CO at a commercially relevant scale (at least $j_{CO} > 200$ mA cm$^{-2}$) due to the partial reduction of the ligands around the atomically dispersed metal sites under a high applied potential[23-26].

Cu possesses excellent activity towards $CO_2$ activation, but the control over the selectivity of $CO_2$RR on Cu is a major challenge given that it can produce at least sixteen different hydrocarbons/oxygenates[27-30]. Generally, during the $CO_2$RR, $CO_2$ molecules first undergo adsorption and activation on surface atoms. Further proton-coupled electron transfer converts $CO_2^*$ into COOH* (further evolves to CO*) or HCOO*, the key intermediates forming CO and HCOOH, respectively[31,32]. On Cu catalysts, proper binding energy to CO* endows Cu with the ability to couple CO* to generate other $C_{2+}$ products (e.g.,

$C_2H_4$[33], $C_2H_5OH$[34], and $C_3H_7OH$[35]). This situation prompted us to consider whether substantially and acutely tuning the local electronic/geometric configuration of Cu could inhibit C-C coupling without reducing intrinsic activity, thus achieving commercially relevant $CO_2$ electrolysis to CO. We reasoned that weakening the binding strength of H* and CO* intermediates, and thus decreasing the CO* coverage, could potentially suppress HER and CO* dimerization while promoting CO generation during the $CO_2$RR, so as to be the solution of the CO activity-selectivity dilemma on Cu. Herein, we report an antimony-copper single-atom alloy ($Sb_1Cu$) catalyst (Fig. 1a) with isolated Sb-Cu interfaces that catalyzed the efficient conversion of $CO_2$-to-CO with an FE of *ca.* 91% at 500 mA cm$^{-2}$.

## Results

The $Sb_1Cu$ catalyst was synthesized via the co-reduction of $Cu^{2+}$ and $Sb^{3+}$ using a $NaBH_4$ solution in an ice bath (see Methods). The morphology of the as-synthesized Cu-Sb catalyst was characterized by transmission electron microscopy (TEM) with sizes ranging from 10 to 20 nm (Supplementary Fig. 1). The powder X-ray diffraction (PXRD) pattern of the as-prepared Cu-Sb catalyst displayed obvious $CuO_x$ diffraction peaks without Sb or Sb oxides signals, excluding the formation of Sb nanoparticles (Supplementary Fig. 2). The formation of

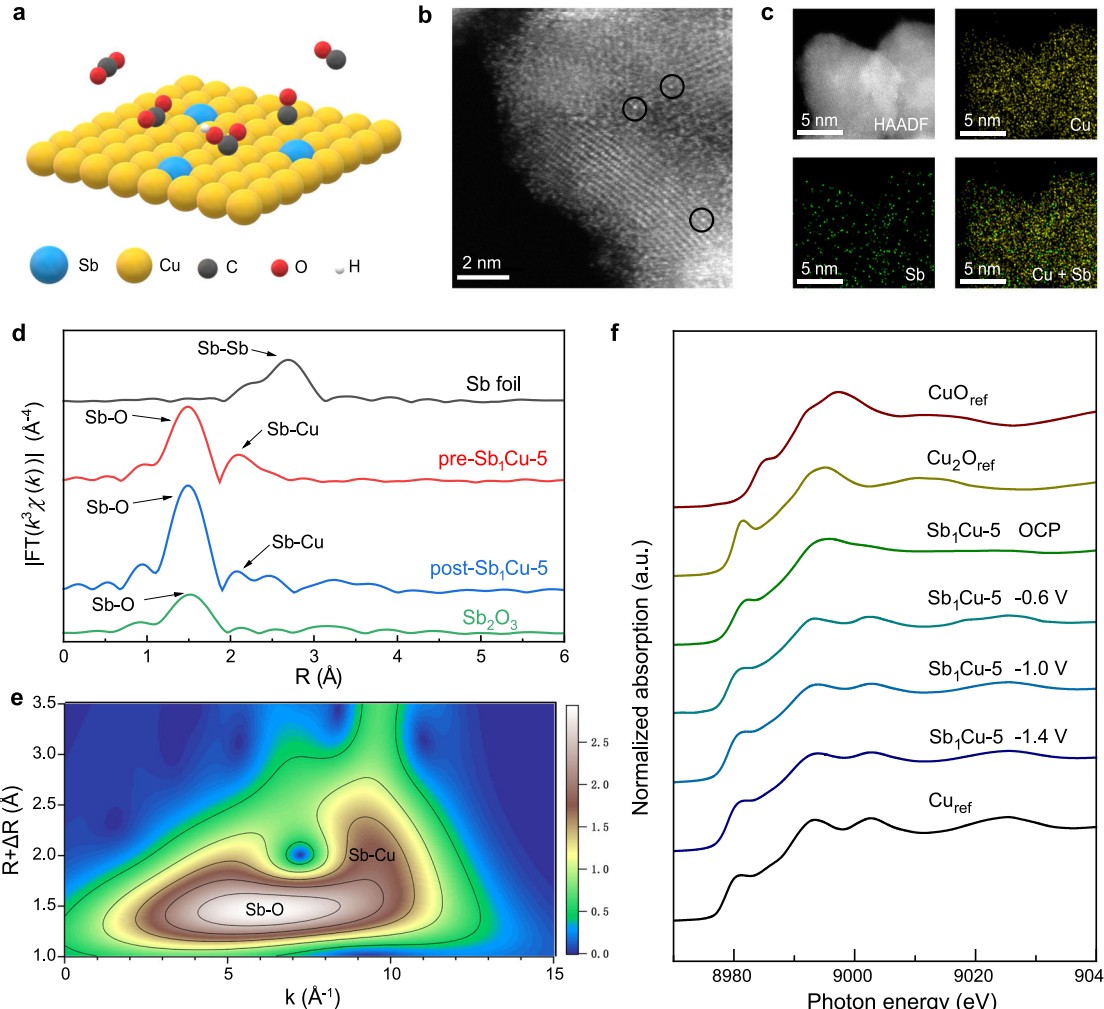

**Fig. 1 | Structural characterization of the $Sb_1Cu$-5 catalyst. a** Schematic illustration of $CO_2$ conversion into CO over the $Sb_1Cu$-5 catalyst. **b** HAADF-STEM image of the as-prepared $Sb_1Cu$-5 catalyst. The black circles highlight single Sb atoms. **c** STEM-EDS mapping of Cu and Sb in the as-prepared $Sb_1Cu$-5 catalyst, confirming the even distribution of Sb in the Cu host. **d** Ex situ EXAFS at the Sb *K*-edge of the as-

prepared and post-catalysis $Sb_1Cu$-5 without phase correction. Sb foil and $Sb_2O_3$ are shown as references. **e** EXAFS wavelet transforms (WT) for the Sb *K*-edge of the $Sb_1Cu$-5 catalyst. **f** In situ Cu *K*-edge XANES spectra of the $Sb_1Cu$-5 catalyst under applied potentials during the $CO_2$RR. Cu, $Cu_2O$, and CuO are shown as references. All potentials were calibrated to the RHE scale.

copper oxides was attributed to the oxygen susceptibility of the Cu nanocrystal surface when exposed to air. The chemical states of $Cu^I$ and $Sb^{III}$ in the Cu-Sb sample were identified by X-ray photoelectron spectroscopy (XPS) (Supplementary Figs. 3–5) and X-ray absorption near-edge structure (XANES) (Supplementary Figs. 6, 7), reconfirming the spontaneous oxidation of the samples. High-angle annular dark-field scanning transmission electron microscopy (HAADF-STEM) clearly identified isolated bright spots representative of atomically dispersed Sb atoms in the Cu matrix for $Sb_1Cu$ (Fig. 1b), suggesting the successful formation of isolated Sb-Cu interfaces. Scanning transmission electron microscopy energy-dispersive X-ray spectroscopy (STEM-EDS) mapping (Fig. 1c) affirmed the even distribution of Sb in Cu without observable Sb aggregation. The concentration of Sb in $Sb_1Cu$ was *ca.* 5.0 at% (denoted as $Sb_1Cu$-5), as determined by inductively coupled plasma atomic emission spectroscopy (ICP-AES). The above analyses implied the formation of a $Sb_1Cu$-5 single-atom alloy with well-defined isolated Sb-Cu atomic sites.

To obtain atomic coordination information of Sb in $Sb_1Cu$-5 samples, extended X-ray absorption fine structure (EXAFS) measurements of the Sb *K*-edge were carried out. As displayed in Fig. 1d, the peak at ~1.50 Å was attributed to the Sb-O bond in the as-synthesized $Sb_1Cu$-5 catalyst. No Sb-Sb bond was observed, indicating the isolated form of Sb atoms. The peak at approximately 2.10 Å was attributed to the Sb-Cu bond, which proved the successful formation of Sb-Cu atomic interfaces. Wavelet transform analysis further corroborated the EXAFS fitting results (Fig. 1e and Supplementary Table 1). The Cu *K*-edge EXAFS results illustrated the coexistence of Cu-Cu bonds (~2.50 Å) and Cu-O bonds (~1.75 Å), indicating the formation of oxides on the surface of the $Sb_1Cu$-5 catalyst (Supplementary Fig. 8), in agreement with the XRD and XPS measurements[32]. We also prepared Cu, $Sb_1Cu$-1.5, and $Sb_1Cu$-10 as controls using a similar strategy (see Methods). Notably, the $Sb_1Cu$-1.5 catalyst presented the same Sb-Cu atomic interface but a lower site concentration than $Sb_1Cu$-5 (Supplementary Figs. 2–12), while the $Sb_1Cu$-10 catalyst showed Sb aggregation (Supplementary Fig. 13). In an effort to elucidate the electronic structure of the $Sb_1Cu$-5 catalyst under reaction conditions, we conducted an in situ X-ray absorption spectroscopy (XAS) study. Cu *K*-edge in situ X-ray absorption fine structure (XAFS) demonstrated that $Cu^I$ in the as-prepared $Sb_1Cu$-5 was reduced to metallic Cu during the $CO_2RR$ (Fig. 1f and Supplementary Fig. 14). Unfortunately, we failed to detect Sb signals in situ, probably due to the strong background interference caused by the fluorescence of Cu and the aqueous environment. Thus, we further explored the atomic geometry of $Sb_1Cu$-5 after $CO_2$ electrolysis. As expected, we still observed highly monodispersed Sb atoms in the Cu host (Supplementary Figs. 15, 16) by HAADF-STEM, in agreement with XRD analysis for post-catalysis $Sb_1Cu$-5 (Supplementary Fig. 17). Moreover, ex situ Sb *K*-edge EXAFS exhibited unchanged isolated Sb-Cu atomic interfaces, suggesting that these atomic sites should be robust during electrolysis (Fig. 1d, Supplementary Fig. 18, and Table 1). The above results, taken together, suggest that the active phase of $Sb_1Cu$-5 during the $CO_2RR$ is metallic Cu alloyed with isolated Sb atoms.

The $CO_2RR$ performance of $Sb_1Cu$-5 was evaluated in a three-electrode flow cell using 0.5 M $KHCO_3$ as the electrolyte (see details in Methods). Gas products were analyzed by gas chromatography (GC), and the liquid products were analyzed using nuclear magnetic resonance (NMR) spectroscopy and anion chromatography (AC). As manifested by the linear sweep voltammetry (LSV) curve of $Sb_1Cu$-5, the current density was much higher under $CO_2$ flow than that in Ar atmosphere, indicating the participation of $CO_2$ in electrolysis (Supplementary Fig. 19). NMR results showed that the solution-phase product only contained formate (Supplementary Fig. 20), and the GC analysis revealed CO and $H_2$ as major gas-phase products (Supplementary Fig. 21). We further found that the $Sb_1Cu$-5 catalyst exhibited high selectivity towards CO even at very high current densities. The

maximal $FE_{CO}$ reached 95% at $-150$ mA cm$^{-2}$, while $FE_{CO}$ remained over 90% even at $-500$ mA cm$^{-2}$ (Fig. 2a). The selectivity towards HER was suppressed to below 3%, while over 90% $FE_{CO}$ was achieved at all applied current densities. The CO partial current density ($j_{CO}$) reached a maximum of $-452$ mA cm$^{-2}$ at $-1.16$ V vs. RHE with 90.4% $FE_{CO}$. In addition, we observed that the $Sb_1Cu$-5 catalyst showed a lower onset potential for $CO_2$-to-CO and much suppressed C-C coupling (>150 mV onset potential for $C_2H_4$) compared to pure Cu using in situ differential electrochemical mass spectrometry (DEMS) (Fig. 2b)[36]. This leads us to believe that the intrinsic activity and selectivity of Cu was significantly modulated by the introduction of isolated Sb-Cu atomic interfaces. The stability test showed that $FE_{CO}$ remained at approximately 95% after 11 h of continuous electrolysis at $-100$ mA cm$^{-2}$ using a membrane electrode assembly (MEA), and the cell voltage was quite stable without obvious fluctuation (Fig. 2c). In order to exclude the influence of structural decay of gas diffusion electrode during long-term electrolysis, we further evaluated the intrinsic stability of $Sb_1Cu$-5 catalyst in an H-cell, exhibiting a stable cathode potential with *ca.* 90% $FE_{CO}$ for 100 h (Supplementary Fig. 22). The impressive performance for $CO_2$-to-CO on $Sb_1Cu$-5 outperforms the previously reported state-of-the-art CO-selective electrocatalysts[37–39] (Fig. 2d, e, Supplementary Fig. 23, and Tables 2–4).

We next analyzed the possible mechanism for exclusive CO production on the $Sb_1Cu$ catalyst. We first compared the $CO_2RR$ performance of pure Cu, Sb, $Sb_1Cu$-1.5, and $Sb_1Cu$-10 catalysts. Various products were obtained in the Cu catalyst, as expected, which showed an increasing $C_{2+}/C_1$ ratio with increasing current density (Fig. 2f). It is worth noting that, compared with $Sb_1Cu$-5 catalyst, Cu exhibited higher selectivity towards $C_{2+}$ products due to the preferable C-C coupling at higher current densities (e.g., 42% at $-400$ mA cm$^{-2}$). The performance of pure Sb is given in Fig. 2g, showing a maximum $FE_{CO}$ of 34.8%. For the $Sb_1Cu$-1.5 catalyst (Fig. 2h), owing to the formation of isolated Sb-Cu atomic interfaces, $FE_{CO}$ reached 89% at $-100$ mA cm$^{-2}$. However, larger amounts of $C_{2+}$ products also appeared at high current densities (up to 27% at $-400$ mA cm$^{-2}$) because only a smaller portion of the pristine Cu surface was modulated by Sb atoms and the coordination environments of most Cu sites remained the same. Owing to the formation of Sb aggregates, a relatively higher selectivity of formate was found on the $Sb_1Cu$-10 catalyst (Supplementary Fig. 24). These comparisons directly link the exclusive CO selectivity to the impact from alloyed Sb single atoms in the $Sb_1Cu$ catalyst.

We then sought to carry out kinetic analysis to further understand how the isolated Sb-Cu atomic interface steers the $CO_2$-to-CO pathway on $Sb_1Cu$-5. Tafel plots (Supplementary Fig. 25) revealed a faster kinetic process of CO generation on the $Sb_1Cu$-5 catalyst (177 mV dec$^{-1}$) compared with pure Cu (253 mV dec$^{-1}$). The Tafel slope of 177 mV dec$^{-1}$ on $Sb_1Cu$-5 indicated that the first electron transfer step of *$CO_2$ was the rate-determining step (RDS)[31]. The slopes were larger than the frequently mentioned Tafel slope of 118 mV dec$^{-1}$ for $CO_2$ activation RDS (Supplementary Table 5), which could be attributed to the differences in the numerical selection for the symmetry factor ($\alpha = 0.5$ for 118 mV dec$^{-1}$) regarding more complicated electron transfer and chemical processes under realistic conditions[40]. Thus, we employed Fourier-transform alternating current voltammetry (FTacV) to gain further mechanistic insights into electron transfer processes during the $CO_2RR$ on $Sb_1Cu$-5[41]. In FTacV, a large amplitude periodic alternating current (ac) waveform was superimposed onto the direct current (dc) ramp to generate higher harmonic components. The high harmonic ac components were highly sensitive to electron transfer kinetics and devoid of nonfaradaic background current contributions[42], thereby leading to direct access to underlying electron transfer processes. The fourth harmonic components of FTacV obtained on $Sb_1Cu$-5 in $CO_2$-saturated 0.5 M $KHCO_3$ showed a well-defined reduction peak at approximately $-0.60$ V vs. RHE,

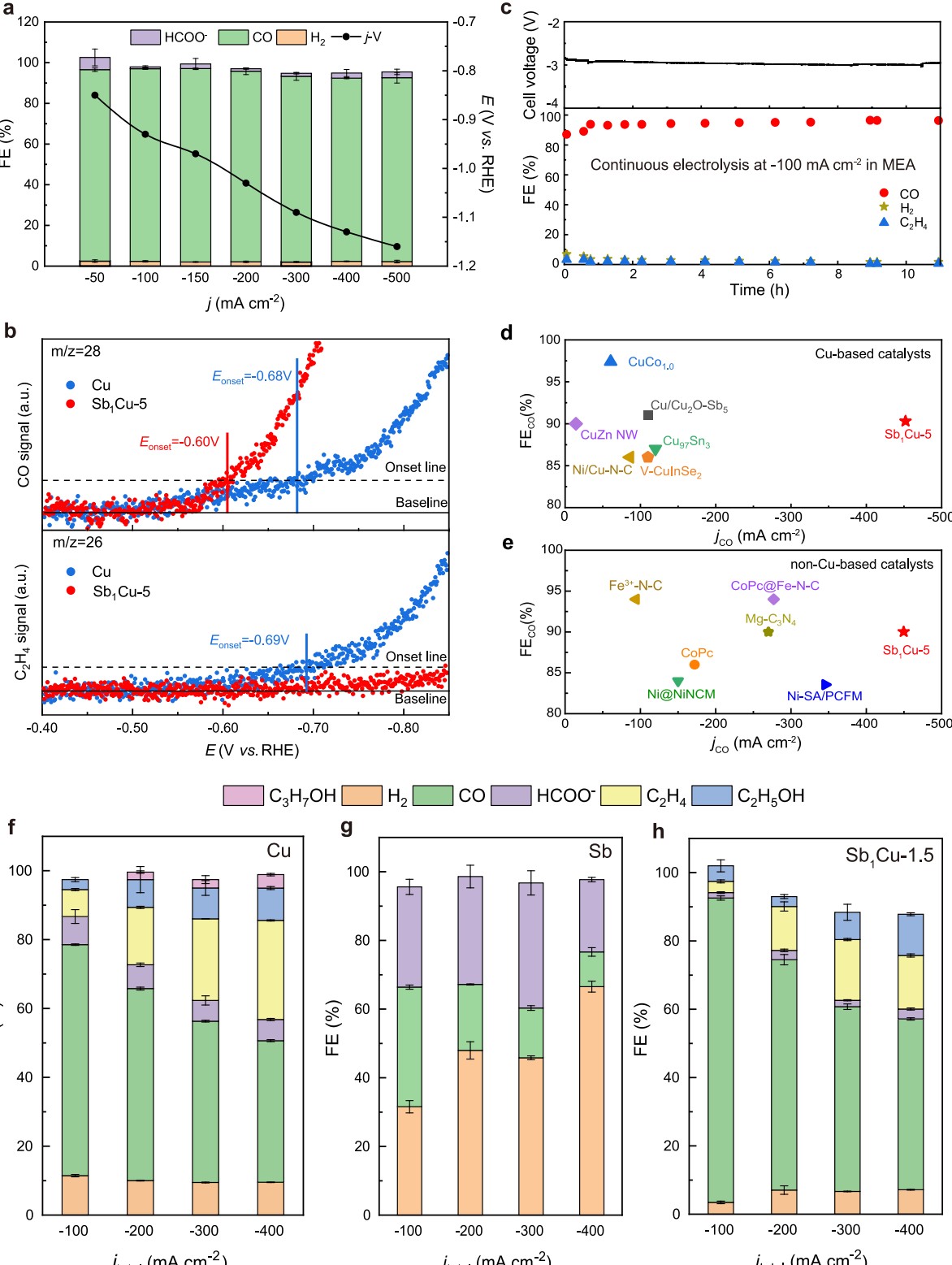

**Fig. 2 | CO₂RR performance over Sb₁Cu catalysts and Cu. a** FEs of all CO₂RR products at different current densities and the corresponding *j*-V curve of Sb₁Cu-5. **b** In situ DEMS measurement of (up) CO and (down) C₂H₄ production during CO₂RR on Cu and Sb₁Cu-5 catalysts. **c** Stability test at −100 mA cm⁻² current density in MEA for over 10 h, indicating an average CO FE of approximately 95%, as estimated by GC analysis. Performance metrics of different reported CO₂RR-to-CO **d** Cu-based and **e** non-Cu-based catalysts. Comparison of product FEs on **f** Cu, **g** Sb, and **h** Sb₁Cu-1.5. The error bars correspond to the standard deviation of three independent measurements.

which was absent in the Ar-saturated electrolyte, confirming the involvement of $CO_2$ in the process (Fig. 3a). Simulations of FTacV (see details in Methods) showed that the standard rate constant ($k_s$) of the one-electron $CO_2$-to-$^*CO_2^-$ the process was smaller than that for the one-electron $^*COOH$-to-$^*COOH^-$ (Supplementary Table 6), indicating the first electron-transfer step as the RDS. In addition, based on the fitting transfer coefficient ($\alpha = 0.34$) for $CO_2$-to-$^*CO_2^-$, we estimated the Tafel slope to be 174 mV dec$^{-1}$ on $Sb_1Cu$-5, which is close to the experimental value of 177 mV dec$^{-1}$ and thus, in turn, verifies our simulations. Taken together, the Tafel and FTacV analyses indicated an improved electron transfer process of CO production on $Sb_1Cu$-5, highlighting the critical role of isolated Sb atoms in boosting $CO_2$-to-CO conversion.

In pursuit of a molecular-level understanding of the $CO_2$-to-CO conversion pathway, we first conducted in situ attenuated total reflection surface-enhanced infrared absorption spectroscopy (ATR-SEIRAS) to dynamically monitor the evolution of reactive intermediates. Figures 3b and 3c show the in situ ATR-SEIRAS spectra acquired for the Cu and $Sb_1Cu$-5 electrodes during a negative-going potential sweep from −0.2 to −0.8 V vs. RHE. Upon applying the cathodic potential, a fingerprint infrared band appeared at 2000–2100 cm$^{-1}$ for both $Sb_1Cu$-5 and Cu catalysts, which was assigned to surface-bond CO (CO*). For either electrode, the CO* band frequencies redshifted with a negatively going potential due to the Stark effect. Notably, the CO* bands are redshifted and broadened on $Sb_1Cu$-5, in comparison to the counterparts on Cu at given potentials. As the surface coverage could impact the frequency of vibrational bands via dynamical dipole coupling[43], the lower frequency of the CO* peak on $Sb_1Cu$-5 implied weakened CO* adsorption and much lower CO coverage relative to Cu (Fig. 3d). After increasing the bias to −0.8 V vs. RHE, the CO* peak vanished on $Sb_1Cu$-5 but was still present on Cu, indicating more facile desorption of the CO intermediate from the $Sb_1Cu$-5 surface to form gaseous CO. In addition, after suspension of the applied potential and sweeping adsorbates with Ar flow, $Sb_1Cu$-5 showed a faster attenuation rate and shorter retention time of CO* compared with pure Cu, further confirming its better CO desorption ability (Supplementary Fig. 26). The same conclusions were also drawn from the in situ Raman spectra, where the peaks at 2000–2100 cm$^{-1}$ were ascribed to CO* intermediates[44] (Fig. 3e). As shown in Fig. 3f, the peak at 1060 cm$^{-1}$, which is associated with $\nu(CO_3^{2-})$ vibration[45], was much weaker on $Sb_1Cu$-5 than that on Cu at a given potential, inferring a surface environment of a lower pH value[46]. It has been demonstrated that a higher surface pH is conducive to promoting C-C coupling[47]. Thus, endowed with the capabilities of weak adsorption for CO* and facile desorption for CO, $Sb_1Cu$-5 predominantly prefers CO evolution to CO* dimerization in the case of lower surface pH.

Density function theoretical (DFT) calculations were performed to further understand the enhanced $FE_{CO}$ on $Sb_1Cu$-5. As discussed previously[48], a step surface is more active for the $CO_2RR$. Thus, the Cu (211) surface model was first constructed (Supplementary Fig. 27). A single Sb dopant was substitutionally created to simulate a $Sb_1Cu$-5 (211) surface. It was found that the electrochemical interface is close to a capacitor (Supplementary Fig. 28) regarding the correlations between the amount of electron transfer from the electrode to the water layer ($\Delta q$) and the relative work function change of the systems ($\Delta\Phi$) at initial, transition, and final states[49]. Therefore, the activation barriers at varying potentials were explicitly calculated within a capacitor model with the "charge-extrapolation" scheme[50–52]. Figure 4a shows that $CO_2$ can be electroreduced to CO and HCOOH via COOH* or HCOO* pathway, respectively, at −1.0 V vs. RHE. The kinetic barrier of HCOOH formation is higher than that for CO production on both Cu (211) and $Sb_1Cu$-5 (211), which is consistent with the experiments that $FE_{HCOOH}$ is lower than $FE_{CO}$ on both Cu and $Sb_1Cu$-5. Figure 4b shows that all the adsorption energies of CO*, COOH*, and

HCOO* on $Sb_1Cu$-5 (211) are lower than those on Cu (211). The CO* desorption is easier on $Sb_1Cu$-5 (211) compared to Cu (211), which can enhance CO production and selectivity. To further understand the effect of isolated Sb in Cu, the projected density of states (PDOS) was calculated for Cu (211) and $Sb_1Cu$-5 (211) (Fig. 4c). The $d$-band center of the Cu atom is lower after Sb doping (from −2.07 to −2.17 eV), which explains well the weaker adsorption energy on $Sb_1Cu$-5 (211) than that for Cu (211).

As $C_{2+}$ products also have a lower FE for the $CO_2RR$ on the $Sb_1Cu$-5 electrode, the barriers of CO*-CO* coupling were calculated. As shown in Fig. 4d, the kinetic barrier of CO* dimerization on $Sb_1Cu$-5 (211) is higher than that on Cu (211), thus resulting in slower production of $C_{2+}$ products, and this is another reason for the high $FE_{CO}$ on $Sb_1Cu$-5. Microkinetic modeling further validated the enhanced $FE_{CO}$ on $Sb_1Cu$-5 (211), and the results are shown in Fig. 4e. The calculated $FE_{CO}$ was comparable to the experimental results for both pristine Cu and $Sb_1Cu$-5. It was found that CO* takes up 70% coverage of the total sites for the $CO_2RR$ on pristine Cu, whereas that on $Sb_1Cu$-5 (211) is only 7% (Supplementary Fig. 29). Finally, we further calculated the activity of CO production on $Sb_1Cu$-5 (211) at varying potentials (Fig. 4f). As the potential decreases, all the barriers of electrochemical protonation gradually decrease, which will increase the reaction rate. A good linear correlation (slope = 1.05) was obtained between the experimental $j_{CO}$ and the theoretical rate of CO production, as shown in Fig. 4g. This indicates that the reaction mechanism and energetic trend above should be reliable.

To sum up, in this work, $Sb_1Cu$ catalysts were successfully synthesized using a co-reduction method, with isolated Sb-Cu atomic interfaces verified by HAADF-STEM and EXAFS. $Sb_1Cu$-5 achieved an $FE_{CO}$ of 95% at a current density of −150 mA cm$^{-2}$ and maintained >90% $FE_{CO}$ at −500 mA cm$^{-2}$. Distinct from Cu with major $C_{2+}$ products at high current density, high activity and selectivity towards CO production on $Sb_1Cu$-5 was elaborated by tuning the Cu electronic structure via forming an isolated Sb-Cu atomic interface, and the improved performance can be explained by the enhanced ability of $CO_2$ activation and CO desorption for Cu. This report provides a facile strategy to manipulate the reaction pathway through isolated heteroatom interfaces, which could be widely used in general element combinations and other electrocatalytic reactions.

## Methods
### Material synthesis
**Chemicals.** Copper (II) chloride dihydrate ($CuCl_2 \cdot 2H_2O$, ACS), antimony trichloride ($SbCl_3$, 99.9%), citric acid ($H_3Cit$, 99.5%), sodium borohydride ($NaBH_4$, 97%), hydrochloric acid (HCl, GR), antimony powder (Sb, 99%) and isopropanol (IPA, 99.5%) were purchased from Macklin. All chemicals were used without further purification.

**Synthesis.** $Sb_1Cu$ catalysts were synthesized by co-reduction of $CuCl_2$ and $SbCl_3$ with $NaBH_4$. $SbCl_3$ mixture solution was first prepared, including 1 mmol $SbCl_3$, 3 mmol $H_3Cit$, and 20 ml deionized (DI) water. In the typical synthesis of the $Sb_1Cu$-5 catalyst, 3.2 ml $SbCl_3$ mixture solution, 2 ml 1 M $CuCl_2 \cdot 2H_2O$ solution, and 1 ml 3 M HCl were added into 13.8 ml DI water to obtain the precursor solution. The above solution was then rapidly added into 10 ml 1 M $NaBH_4$ solution in an ice bath and aged for 1 h. After a violent reaction, the obtained black precipitate was then washed with DI water and IPA several times, and dried under a vacuum overnight. The obtained black powder was kept in a glove box under an Ar atmosphere for storage. For the synthesis of the $Sb_1Cu$-1.5 catalyst, 1.2 ml $SbCl_3$ mixture solution, 2 ml 1 M $CuCl_2 \cdot 2H_2O$ solution, and 1 ml 3 M HCl were added into 15.8 ml DI water to obtain the precursor solution. The following steps were the same as $Sb_1Cu$-5.

For the synthesis of the Cu catalyst, the precursor solution was obtained by adding 2 ml 1 M $CuCl_2 \cdot 2H_2O$ solution and 1 ml 3 M HCl into

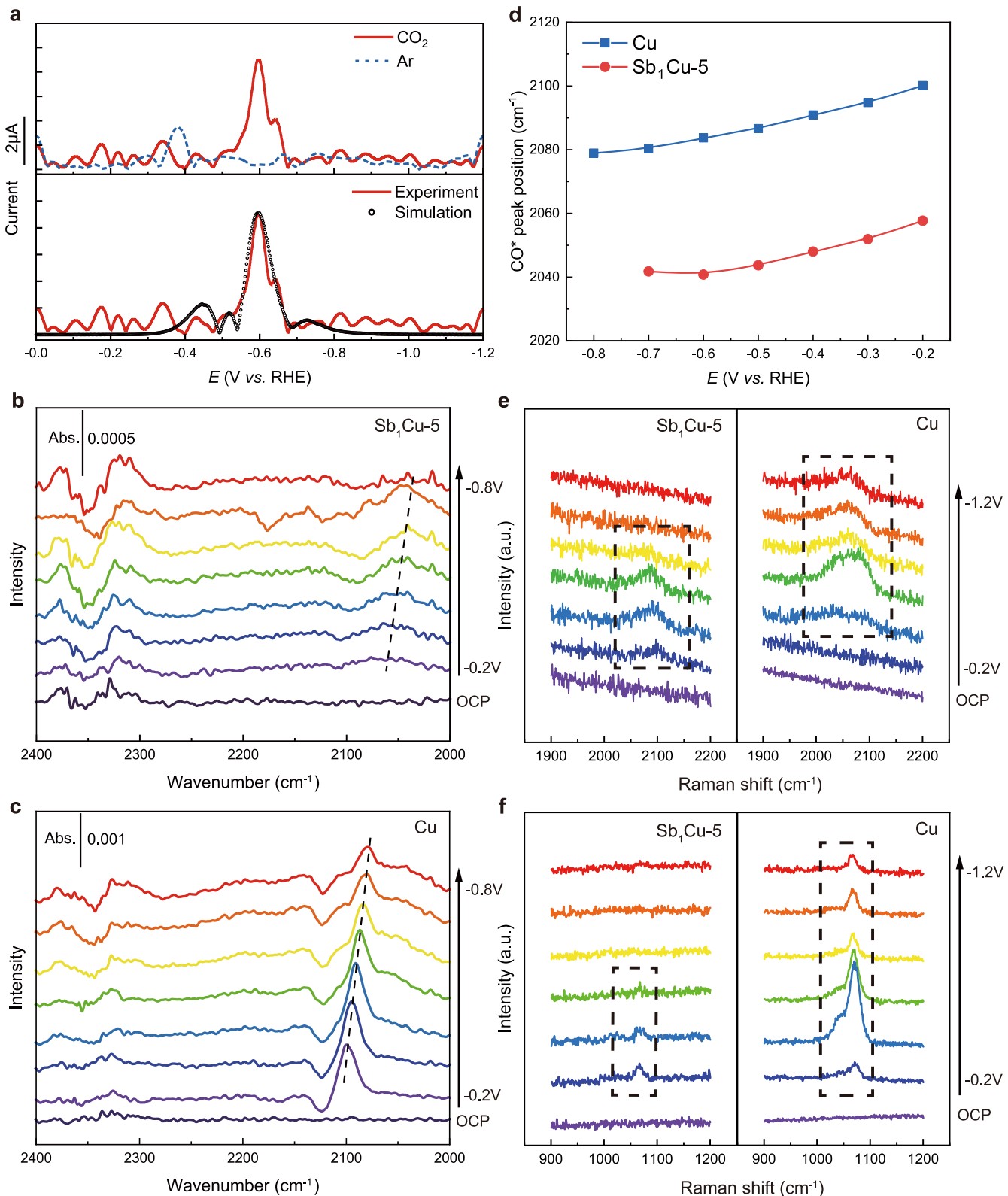

**Fig. 3 | Mechanistic studies of the electrochemical CO₂-to-CO conversion on the Sb₁Cu catalyst. a** Fourth harmonic components of FTacV derived on Sb₁Cu-5 in 0.5 M KHCO₃ solution saturated with CO₂ (solid line) and Ar (dashed line). The FTacV parameters are $f$ = 9.02 Hz, $\Delta E$ = 80 mV, and scan rate = 10.57 mV s⁻¹. The fitting of the fourth harmonic component is also displayed at the bottom. In situ

ATR-SEIRAS spectra of **b** Sb₁Cu-5 and **c** Cu at different potentials. Abs absorbance. **d** Potential-dependent CO* peak position on Sb₁Cu-5 and Cu in in situ ATR-SEIRAS spectra. **e** CO* and **f** CO₃²⁻ peaks of Sb₁Cu-5 and Cu in in situ Raman spectra at different potentials. All potentials were calibrated to the RHE scale.

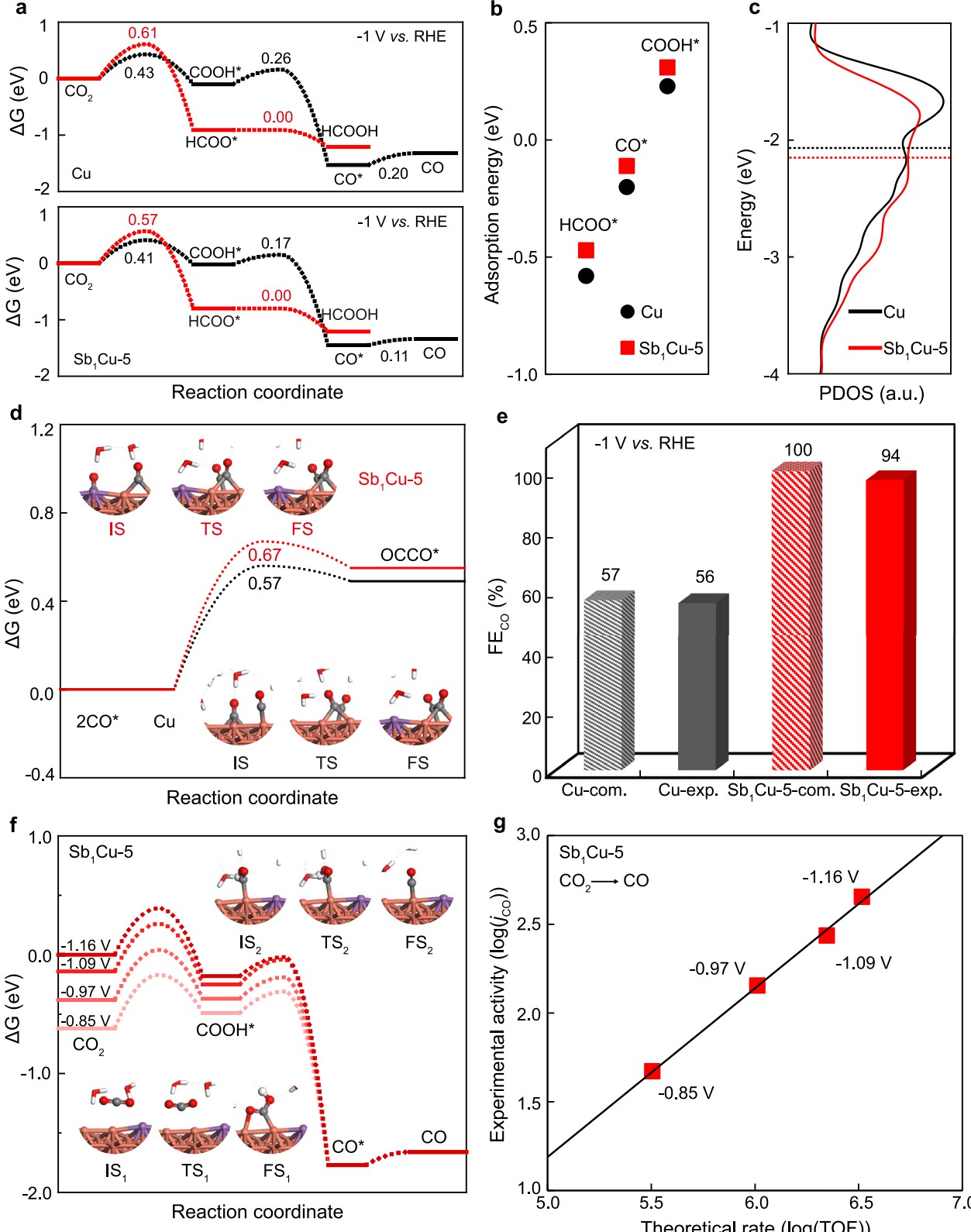

**Fig. 4 | Theoretical calculations. a** $CO_2$RR to CO and HCOOH on Cu (211) and Sb₁Cu-5 (211). **b** The adsorption energies of HCOO*, CO* and COOH* on Cu (211) and Sb₁Cu-5 (211). **c** The projected d-band states of Cu atoms on Cu (211) and Sb₁Cu-5 (211). **d** Barriers for CO*-CO* coupling on Cu (211) and Sb₁Cu-5 (211). The initial (IS), transition (TS), and final (FS) structures are shown as insets, where Cu, Sb, C, O, and H are represented in orange, purple, gray, red, and white, respectively.

**e** Comparison between the calculated $FE_{CO}$ from microkinetic simulations on Cu (211) and Sb₁Cu-5 (211) and the experimental $FE_{CO}$ on Cu and Sb₁Cu-5. **f** Free energy diagrams of the $CO_2$RR into CO under different potentials; insets are structures of the initial (IS), transition (TS), and final (FS) states of $CO_2$ protonation (1) and CO* formation (2). The symbols with the same color represent the same atoms in figure **d** and **f**. **g** Experimental activity versus a theoretical rate of CO production.

17 ml DI water. For the synthesis of the Sb$_1$Cu-10 catalyst, the precursor solution was obtained by adding 8 ml SbCl$_3$ mixture solution, 2 ml 1 M CuCl$_2$·2H$_2$O solution, and 1 ml 3 M HCl into 9 ml DI water. The following steps were the same as the synthesis of Sb$_1$Cu.

## Electrochemical measurements

**CO$_2$RR performance test.** All electrochemical measurements were conducted at room temperature using BioLogic VMP3. Typical three-electrode cell measurements were performed using a conventional flow cell and a glass H-cell. For the CO$_2$RR measurements in the flow cell, precursor ink (18 mg of catalyst, mixed with 36 μL of 5% Nafion 117 solution solved in 5 ml of IPA) was sprayed onto a gas diffusion layer (YLS-30T) as the cathode electrode with a mass loading of ~1 mg cm$^{-2}$. The Ag/AgCl wire in saturated KCl solution was used as the reference electrode, and Ni foam was used as the counter electrode. The working and counter electrodes were then placed on opposite sides of two 1-cm-thick polytetrafluoroethylene (PTFE) sheets with 0.4 cm × 1.5 cm channels such that the catalyst layer interfaced with the flowing electrolyte. The geometric surface area of the catalyst was 0.6 cm$^2$. A Nafion 115 membrane was sandwiched between the two PTFE sheets to separate the chambers. CO$_2$ flowed through the gas room behind the cathode, and the flow rate was kept at 30 sccm (monitored by an Alicat Scientific mass flow controller). In addition, 0.5 M KHCO$_3$ was circulated as the cathode electrolyte at a flow rate of 0.7 ml min$^{-1}$, while 1 M KOH was purged as the anode electrolyte. For measurements in the H-cell, catalysts were loaded on carbon paper (1 cm × 2 cm) as the working electrode. 0.5 M KHCO$_3$ as the cathode electrolyte and 1 M KOH as the anode electrolyte were separated by a Nafion 115 film. All potentials were converted to the RHE reference scale using the relation $E_{RHE} = E_{Ag/AgCl} + 0.197 + pH × 0.059$ and compensated for the solution resistance by 85%.

**CO$_2$RR product analysis.** The gaseous products were tested by an online GC (PerkinElmer Clarus 690), which was equipped with a flame ionization detector, a thermal conductivity detector, and Molsieve 5 Å Column. The liquid products of Cu and Sb$_1$Cu-1.5 were quantified by a 400 MHz NMR spectrometer (BUKER). About 100 μl of D$_2$O (Sigma Aldrich, 99.9 at.%) and 0.05 μl of dimethyl sulfoxide (Sigma Aldrich, 99.9%) as an internal standard was added into 600 μl of the electrolyte after the electrolysis. The liquid products of Sb, Sb$_1$Cu-5, and Sb$_1$Cu-10 were determined by NMR and ion chromatography (Thermo Fisher Scientific ICS-600).

## Long-term stability test

A membrane electrode assembly (MEA) was used for the long-term stability test with a zero-gap configuration where the anode, membrane, and cathode were compressed together to form one reactor. IrO$_2$/Ti mesh was used as the anode, and an anion exchange membrane (Sustainion X37-FA, Dioxide Materials) was placed between the anode and cathode. CO$_2$ was humidified upstream to the MEA and fed to the GDL cathode at 50 sccm. About 0.1 M KHCO$_3$ was fed to the anode at 3 ml min$^{-1}$ to supply anode oxygen evolution. A 100-h intrinsic stability test was conducted in the H-cell. Catalysts were loaded on carbon paper (1.5 cm × 2 cm) as the working electrode, with a graphite rod as the counter electrode. 0.5 M KHCO$_3$ as the cathode electrolyte and 1 M KOH as the anode electrolyte were separated by a bipolar film. A 20 sccm CO$_2$ flow was bubbled into the cathode electrolyte.

## Tafel plot

The Tafel plots were employed to evaluate the CO$_2$RR catalytic kinetics and fitted with the following equation: $η = k × \log(j_{CO}) + b$, where $j_{CO}$ is the CO partial current density, η is the overpotential for CO$_2$+H$_2$O+2e$^-$ →CO+2OH$^-$ ($E^0 = -0.11$ V vs. RHE). The smaller slope k indicates faster kinetics towards CO production. If the rate-determining step (RDS) is the first electron transfer CO$_2$-to-*CO$_2^-$, the Tafel slope is calculated by the following formula:

$$\frac{\partial(-η)}{\partial \lg(j_{CO})} = \frac{2.3RT}{αF} \qquad (1)$$

In this equation, $α$ is the transfer coefficient and $F$ is the Faraday constant. The standard values of the Tafel slopes based on different RDSs are further given in Supplementary Table 5.

## FTacV measurements and analysis

FTacV experiments were carried out by a CHI 660e electrochemical workstation using an applied sine wave perturbation with an amplitude ($ΔE$) of 80 mV and a frequency ($f$) of 9.02 Hz superimposed onto the direct current (dc) ramp from 0 V to −1.20 V vs. RHE with a scan rate ($v$) of 10.57 mV s$^{-1}$. A standard three-electrode H-cell setup was used. Ni foam was used as the counter electrode, and Ag/AgCl was used as the reference electrode. Sb$_1$Cu-5 catalyst was coated on glassy carbon as the working electrode. After the use of an FT-inverse FT sequence, the ac data can be resolved into a dc component as well as fundamental, second, and higher-order ac harmonic components. The fourth harmonic components are very sensitive to fast heterogeneous electron transfer processes but essentially devoid of contributions from catalytic and background charging currents. The MECSim package was used for the simulation of the fourth harmonic component in FTacV, utilizing the ECEC model for four elementary steps, as illustrated in Supplementary Table 6. The parameters of kinetics were fitted through trial and error until a good fit was achieved.

## In situ DEMS measurement

In situ differential electrochemical mass spectrometry (DEMS) was performed using a custom-made electrochemical capillary DEMS flow cell. The catalysts were loaded on the gas diffusion layer as the cathode, where CO$_2$ flowed behind. A capillary was put into the gas outlet of the flow cell to draw the gas products into the DEMS sensor (PrismaPro). The signals of the mass-to-charge ratio (m/z) of 26 and 28 represented the products of C$_2$H$_4$ and CO, respectively. Linear sweep voltammetry (LSV) with a scan rate of 5 mV s$^{-1}$ was conducted on the cathode. The onset potentials were determined according to the positions where the signal-to-noise ratio was greater than 5.

## Characterization techniques

Transmission electron microscope (TEM) images were conducted on Hitachi H-7650 TEM equipment, and the acceleration voltage was 100 kV. Powder X-ray diffraction (PXRD) patterns were taken on a Philips X'Pert Pro Super diffractometer, and the standard λ value was 1.54178 Å for Cu-Kα radiation. X-ray photoelectron spectroscopy (XPS) measurements were conducted on VG ESCALAB MK II equipment, and the exciting source was Al Kα = 1486.6 eV. C 1 s of 284.6 eV was used to reference the binding energies. HAADF-STEM images and energy-dispersive spectra (EDS) elemental mapping were carried out on a Themis Z field-emission transmission electron microscope using Mo-based TEM grids, and the accelerating voltage was 200 kV. The in situ X-ray absorption fine structure (XAFS) spectra of the Cu K-edge were obtained at the BL11B beamline of the Shanghai Synchrotron Radiation Facility with a constant current of 200 mA, recorded under fluorescence mode with a Lytle detector in an H-cell, operated at 3.5 GeV under "top-up" mode. The XAFS spectra of the Sb K-edge were obtained at beamline 44 A of the Taiwan Photon Source (TPS). The Demeter software package was used to process the XAFS data. In situ Raman analysis was conducted using a LabRAM HR laser Raman analyzer (Horiba/Jobin Yvon, Longjumeau) equipped with a frequency-doubled Nd:YAG 785 nm laser. In situ electrochemical attenuated total reflection surface-enhanced infrared absorption spectroscopy (ATR-SEIRAS) experiments were conducted on a Thermo Scientific Nicolet

iS50 FTIR spectrometer with silicon as the prismatic window at room temperature.

## Computational details

We employed the Vienna Ab initio Simulation Package (VASP) to perform all density functional theory (DFT)[53,54] calculations with the generalized gradient approximation (GGA) using the revised Perdew–Burke–Ernzerhof (rPBE) functional[55]. We chose the projected augmented wave (PAW)[56,57] and a plane wave basis set with a kinetic energy cutoff of 400 eV. Geometry optimizations were performed with a force convergence smaller than 0.05 eV Å$^{-1}$. Cu (211) and Sb$_1$Cu-5 (211) surface models were built with four layers comprising 48 atoms. The two layers at the bottom were fixed, while the other atoms relaxed. Monkhorst-Pack $k$-points of ($4 \times 2 \times 1$) were applied for all the calculations on Cu (211) and Sb$_1$Cu-5 (211). The Cu site nearest to the Sb atom has been discussed in the main text. The Cu sites far from the Sb atom should be very similar to pure Cu. We further studied the reaction activity of the Cu site, which is the next-nearest to the Sb atom on Sb$_1$Cu-5 (211), as shown in Supplementary Fig. 30. CO is also the main product due to the lower barrier compared to HCOOH and C$_{2+}$ formation. However, the adsorption energy of CO* at the top site on Cu (next-nearest to Sb atom, −0.21 eV) was close to pure Cu (−0.20 eV), more stable than CO* at the top site of Cu (nearest to Sb atom, −0.11 eV). CO* accounts for 77% of the total sites for CO$_2$RR. This is inconsistent with the results of spectroscopic measurements, where the lower frequency of the CO* peak on Sb$_1$Cu-5 implied weakened CO* adsorption and much lower *CO coverage relative to Cu (Fig. 3d). Hence, the Cu sites (next-nearest to Sb atom) on Sb$_1$Cu-5 should not be the main active sites, as listed in the supplementary information. The Cu sites (nearest to the Sb atom) on Sb$_1$Cu-5 should have major activity contributions, as shown in the main text.

All the adsorption energies were referenced to the gas phase energies of CO, H$_2$O, and H$_2$. The reaction-free energies ($\Delta G$) were calculated as follows. $\Delta G = \Delta E + \Delta ZPE - T\Delta S$ ($T = 300$ K), where $\Delta E$ is the electronic energy based on DFT calculations directly, and $\Delta ZPE$ and $\Delta S$ are the corrections of zero point energy and entropy, respectively. Only vibrational motion was considered for adsorbates on the surface, while translational, rotational, and vibrational motions were all calculated for gas-phase species. The climbing image nudged elastic band (CI-NEB) method was used to locate the transition states[58]. The solvation effect was also calculated using implicit models through VASPsol[59] calculations for both Cu (211) and Sb$_1$Cu-5 (211). It has been validated that the CO adsorption energy on Cu (211) is comparable with previous work[48] with an explicit solvation model and van der Waals corrections. In addition, the chemical potential of (H$^+$ + e$^-$) was calculated by G (H$^+$ + e$^-$) = ½ G (H$_2$) at 0 V vs. RHE. A computational hydrogen electrode model was used to calculate the free energy change at varying potential[60]. Note that the desorption of CO* was considered to be potentially independent in the calculation of free energy change[32].

Microkinetic modeling was used to simulate the reaction rate in the CO$_2$RR, solved by CATKINAS code:[61,62]

$$\frac{\partial \theta_i}{\partial t} = 0 \tag{2}$$

$$\sum_i \theta_i = 1 \tag{3}$$

The reaction rate on surfaces was described by[63]

$$r = \theta_A \theta_B \frac{k_B T}{h} e^{-G_a/k_B T} \tag{4}$$

Equations (2–4) were referred to in a previous work[64]. A temperature of 300 K was applied with all the gas-phase pressures (including $P_{H^+}$) set to 1.0[65]. The reaction rate of CO$_2$RR to C$_{2+}$ products was estimated based on the Arrhenius equation and the CO* coverage at a steady state.

The FE was described by

$$\text{FE}/\% = \frac{n(i)\text{TOF}(i)}{\sum n(i)\text{TOF}(i)} \times 100 \tag{5}$$

where n($i$) represents the electron transfer number and TOF($i$) is the turnover frequency obtained by microkinetic simulation for product $i$.

We have also strictly investigated the potential effects on the adsorption energy of CO*. An electric field was applied in DFT calculations. Based on a parallel-plate capacitor model, a linear correlation between the electric field and absolute potential was approximated as following:[66]

$$E = \frac{\sigma}{\varepsilon \varepsilon_0} = \frac{C_H(U_{SHE} - U_{PZC})}{\varepsilon \varepsilon_0} \tag{6}$$

where $\sigma$ is the surface charge density and $\varepsilon$ and $\varepsilon_0$ are the dielectric constants of vacuum and water near the interface, which were set to $8.85 \times 10^{-12}$ F m$^{-1}$ and 2 (unitless), respectively. $C_H$ refers to the Helmholtz capacitance ($\mu$F cm$^{-2}$), which was set to 25 $\mu$F cm$^{-2}$. $U_{SHE}$ is the electrode potential referenced to a standard hydrogen electrode (SHE). $U_{PZC}$ refers to the potential of zero charge (PZC) versus SHE, which was set to be −0.9 V[67].

The SHE could be converted to the RHE by the following formula:

$$U_{RHE} = U_{SHE} + 0.059\text{pH} \tag{7}$$

The calculated adsorption energies of CO* changed very little (<0.05 eV) with potentials between 0 and −1.2 V vs. RHE, which showed consistent trends for Cu (211) and Sb$_1$Cu-5 (211). All insights and conclusions shown in the main text are still reliable. In addition, by varying potentials from −0.2 to −0.7 V vs. RHE, the calculated adsorption energies of CO* weaken from −0.13 to −0.11 eV on Sb$_1$Cu-5 (211) and from −0.23 to −0.20 eV on Cu (211), which showed consistent trends with the results of the in situ ATR-SEIRAS spectra (Fig. 3b–d).

Additional microkinetic modeling was performed to double-check the reliability of the present kinetic analysis. All conclusions are not affected. Overall, the potential effects on the adsorption energy of CO* were ignored in the calculation of free energy change.

## Data availability

All data were available in the main text or the supplementary materials. Source data of the figures in the main text are provided. Source data are provided with this paper.

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

## Acknowledgements

C.X. acknowledges the National Key Research and Development Program of China (2022YFB4102000), NSFC (22102018 and 52171201), the Natural Science Foundation of Sichuan Province (2022NSFSC0194), the Central Government Funds of Guiding Local Scientific and Technological Development for Sichuan Province (2021ZYD0043), the Hefei National Research Center for Physical Sciences at the Microscale (KF2021005), and the University of Electronic Science and Technology of China for startup funding (A1098531023601264). J.Z. acknowledges the National Key Research and Development Program of China (2021YFA1500500 and 2019YFA0405600), the National Science Fund for Distinguished Young Scholars (21925204), the National Natural Science Foundation of China (NSFC; U19A2015), the Dalian National Laboratory (DNL) Cooperation Fund, Chinese Academy of Science (CAS; DNL202003), K.C. Wong Education (GJTD-2020-15), Fundamental Research Funds for the Central Universities, Provincial Key Research and Development Program of Anhui (202004a05020074) and University of Science and Technology of China (USTC) Research Funds of the Double First-Class Initiative (YD2340002002). J.X. acknowledges the National Key Research and Development Program of China (No. 2021YFA1500702), the NSFC (22172156), the DNL Cooperation Fund, CAS (DNL202003), the AI S&T Program of Yulin Branch, Dalian National Laboratory For Clean Energy, CAS, Grant No.DNL-YLA202205, and the Strategic Priority Research Program of the Chinese Academy of Sciences (XDB36030200). T.Z. acknowledges the NSFC (22005291 and 22278067). We thank beamline BL11B of the Shanghai Synchrotron Radiation Facility and 44 A of the Taiwan Photon Source (TPS) at the National Synchrotron Radiation Research Center, Taiwan, for providing the beamtime.

## Author contributions

C.X. conceptualized the project. C.X., J.Z., and J.X. supervised the project. J.L. and H.Z. prepared the catalysts and performed the catalytic tests. J.L., H.Z., Z.X., D.Z., and T.Z. performed the catalyst characterizations. S.H. conducted the HAADF-STEM characterizations. Y. Dai and P.C. performed the EXAFS measurements. J.L. and R.Z. performed the in situ measurements. X.D., J.X., Y. Ding, and L.Z. performed the DFT calculations. J.L., C.X., J.Z., and J.X. wrote the paper with input from all authors. All authors discussed the results and commented on the manuscript.

## Competing interests

A China provisional patent application (CN202210385531.1) based on the technology described in this work was filed in April 2022 by C.X., J.L., and H.Z. at the University of Electronic Science and Technology of China. The remaining authors declare no competing interests.
