## [Peer Review File · Nature Communications]

REVIEWER COMMENTS

Reviewer #1 (Remarks to the Author):

In this work, the authors report an antimony-copper single-atom alloy catalyst with isolated Sb-Cu interface for electrochemical CO₂ reduction to CO. The authors attribute the decent performance to that atomic Sb-Cu interface promoted the CO₂ adsorption/activation and weakens the binding strength of CO*. I consider this to be a nice piece of research which will be of interest to the readership of Nature communication. I would like to discuss some aspect before to accept the manuscript.

- (1) According to the results of EXAFS, it seems that the coordination number of Sb-O bond is increased after reaction. It is suggested to fit the results and discuss whether the coordination structure of the catalyst changes significantly after reaction.
- (2) In the reviewer's opinion, the current stability data is not enough to support the conclusion regarding the catalyst with impressive performance rather than performing economically at scale. It is strongly recommended to extend the duration of stability test and further explore the structure evolution of catalysts (such as the coordination structure of Sb).
- (3) Expect for the activity, the selectivity of Cu was also modulated by the introduction of Sb. Thus, the discussion about catalytic performance should be modified.
- (4) Additionally, catalytic performance of the Cu-Bi random alloy catalysts are very essential, which would be much helpful to reveal or confirm the role of isolated Sb-Cu atomic interface.
- (5) Since the efficient three-phase contact of CO₂ (gas), H₂O (liquid) and catalyst (solid) is of great importance for the performance, what about mass transfer?
- (6) It is suggested to give a radar chart when comparing with the previously reported state-of-the-art CO-selective electrocatalysts, which would be much easier to show the advantage.
- (7) The experimental section needs to be far more detailed. It's far too limited for the work to be reproduced effectively. Please can the authors also include a chemicals (source and purity) section.
- (8) Concerning the CO₂-TPD results, whether the desorption peak of could be directly attributed to CO₂? Since the pretreatment temperature is only up to 400 oC, the peak around 400-500 oC needs to be determined in combination with MS results.
- (9) It is suggested to add the results of CO desorption experiment.
- (10) The authors stated that "...the formation of copper oxides was attributed to oxygen susceptibility of the Cu nanocrystal surface when exposed to air". Then, how to guarantee that samples prepared from different batches are consistent. Whether this factor affect the catalytic performance?

Reviewer #2 (Remarks to the Author):

The authors study a copper electrocatalyst doped with slight amounts of Sb for the electrochemical reduction of CO₂ to CO. There are already many reports out on Cu based multimetallic systems for CO₂ electroreduction. Therefore, the approach the authors have taken is not extremely novel. The results are however interesting, as they show that CO can be produced selectively over a range of conditions, including high current density measurements in a MEA setup. The study mentions 1.5 and 5 % Sb in Cu, but does not mention (or investigate) what the optimal composition would be. Apparently there is a lower limit of Sb for the system to be selective to CO. Is there a upper limit for Sb concentration? Overall, the authors perform a very thorough investigation, with detailed material characterization, spectroelectrochemistry and DFT modeling. In general, I think this study can be considered for publication in Nature Communications.

Reviewer #3 (Remarks to the Author):

In this manuscript, the authors report that the Sb-Cu single-atom alloy catalyst constructed by a simple and mild method shows a high CO selectivity for the CO₂ reduction (CO₂RR) with a FE over 95%. Furthermore, a current density of 452 mA/cm² is realized in a designed flow cell. By combining online mass spectrometry, in-situ vibration spectroscopy and theoretical calculation, the high CO selectivity on Sb-Cu catalyst was also explored and explained. The authors propose that the isolated Sb-Cu interface is mainly accountable for the high CO selectivity. However, there have been a number of reports regarding Cu-Sb catalyzed electroreduction of CO₂ to CO with high selectivities despite that the authors ignore others' work for unknown reason. Considering these, the novelty of this paper is incremental but not substantial, therefore, in my opinion, this paper needs improvement before publication. Moreover, there are some other issues that should be addressed.

1. How to define single-atom alloy catalysts? Traditionally, single-atom and alloy are two concepts. In Fig. 1, how to distinguish Sb single-atom from surface Sb-Cu single-atom alloy?
2. At present, the optimal Sb loading mentioned in the article is about 5%. Is it possible for the Sb loading to be increased significantly so that more Sb-Cu interfaces can be constructed in favor of CO₂ to CO conversion?
3. The authors are suggested to run the CO-TPR experiment in addition to the CO₂-TPR.
4. As Fig. 4f, Supplementary Fig. 22, and spectroscopy experimental, the CO is adsorption the top site of Cu site. And TS geometrical structure of C-C coupling is deviated the site, obviously. The authors are suggested to inset the IS and FS structures in Fig. 4d not only TS. And the reaction activity of different Cu

site which close or far Sb atom can further discuss to explain the adsorption site of CO and other intermediates.

5. The detail reaction free energies are calculated, while, in Fig. 4f, the authors need explain the reason of the greatly geometrical structures different between FS1 and IS2, it not accomplishes of one structure for COOH* intermediate at this reaction?

6. The obvious Stark slope of CO is obvious, why not the adsorption energy of CO* is not influenced under different potentials at Fig. 4f.

7. The authors are suggested to compare the relevant work in the literature with their own.

8. Some other minor points:

Check the typos throughout the manuscript, e.g., Line 107, NaHB4.

The label of Figure S20 is messed up, a is Cu, b is Sb-Cu?

Check the format of references, especially journal abbreviations.

The Sb₁Cu₋₅, and the Sb₁Cu_{-1.5}, What does subscript 1 mean?

Reviewer 1

In this work, the authors report an antimony-copper single-atom alloy catalyst with isolated Sb-Cu interface for electrochemical CO₂ reduction to CO. The authors attribute the decent performance to that atomic Sb-Cu interface promoted the CO₂ adsorption/activation and weakens the binding strength of CO. I consider this to be a nice piece of research which will be of interest to the readership of Nature communication. I would like to discuss some aspect before to accept the manuscript.*

Response

We highly appreciate the reviewer's high evaluation of our study, as well as the important suggestions that have greatly improved our work.

Comment 1

According to the results of EXAFS, it seems that the coordination number of Sb-O bond is increased after reaction. It is suggested to fit the results and discuss whether the coordination structure of the catalyst changes significantly after reaction.

Response

We thank the reviewer for his/her important suggestion. We have provided the fitting results of Sb K-edge EXAFS in our original supplementary information (**Supplementary Table 1**, copied below). According to the fitting results, the coordination number of the Sb-O bond increased from 4.8 to 5.9 in the *ex situ* test after CO₂ electrolysis. However, combined with the fact that approximately 20% error bound exists in fitting the coordination number [*Nat. Commun.* **10**, 5812 (2019)], we believe that the coordination structure remained almost unchanged after the reaction.

Supplementary Table 1. EXAFS fitting parameters at the Sb K-edge for Sb₁Cu samples before and after the CO₂RR ($S_0^2=0.938$).

Sample	Shell	N^a	$R(\text{Å})^b$	$\sigma^2(\text{Å}^2)^c$	$\Delta E_0(\text{eV})^d$	R factor
Sb foil	Sb-Sb	3	2.91	0.0039	8.3	0.0045
pre-Sb ₁ Cu-1.5	Sb-O	5.4	1.98	0.0022	8.0	0.0015
	Sb-Cu	0.7	2.68	0.0071		
pre-Sb ₁ Cu-5	Sb-O	4.8	1.98	0.0029	7.9	0.0015
	Sb-Cu	2.4	2.64	0.0102		
post-Sb ₁ Cu-1.5	Sb-O	5.8	1.99	0.0037	8.3	0.0039
	Sb-Cu	0.9	2.68	0.0068		
post-Sb ₁ Cu-5	Sb-O	5.9	1.98	0.0018	9.9	0.0007
	Sb-Cu	1.0	2.58	0.0087		

^a N : coordination numbers; ^b R : bond distance; ^c σ^2 : Debye-Waller factors; ^d ΔE_0 : inner potential correction. R factor: goodness of fit. S_0^2 was set to 0.938 for Sb, according to the experimental EXAFS fit of the Sb foil reference by fixing CN as the known crystallographic value.

Comment 2

In the reviewer's opinion, the current stability data is not enough to support the conclusion regarding the catalyst with impressive performance rather than performing economically at

scale. It is strongly recommended to extend the duration of stability test and further explore the structure evolution of catalysts (such as the coordination structure of Sb).

Response

We thank the reviewer for this important suggestion. In our case, the stability test in MEA lasted 10 hours due to flooding and carbonation problems in the gas diffusion electrode (GDE) rather than the degeneration of the catalysts. After long-term CO₂ electrolysis, the loss of hydrophobicity will damage the triple-phase boundary (where CO₂ reduction mainly occurs), obstructing the diffusion pathways for CO₂ and further leading to a sharp decrease in the CO₂RR activity [ChemSusChem **13**, 400-411 (2020)]. In addition, during CO₂ reduction, basification due to hydroxide formation led to the conversion of CO₂ to carbonate, which caused precipitation of salts on the GDE [ChemElectroChem **6**, 5596-5602 (2019)]. This phenomenon will lead to blocking of diffusion passage and greatly influence the stability of the CO₂ reduction reactor (**Fig. R1**). To confirm the intrinsic stability of the Sb₁Cu-5 catalysts, we used an H-cell as the reaction reactor to exclude the factor of an unstable triple-phase boundary. The long-term stability test showed that FE_{CO} remained at approximately 90% throughout 100 hours of continuous electrolysis at a current density of -10 mA cm⁻², and the cathode potential was quite stable without obvious fluctuation (**Supplementary Fig. 22**, copied below). In addition, *ex situ* Sb K-edge EXAFS of the catalyst after 100 hours of CO₂ electrolysis demonstrated intact isolated Sb-Cu interfaces (**Supplementary Fig. 18**, copied below). In summary, the Sb-Cu atomic sites and CO₂-to-CO performance were robust during long-term CO₂RR, indicating potential industrial application of the Sb₁Cu catalysts.

Fig. R1 | Reaction mechanism of carbonate formation during the CO₂RR [Adv. Mater. 2103900 (2021)].

Supplementary Fig. 22 | Stability test at a current density of -10 mA cm^{-2} in an H-cell for 100 h. To explore the intrinsic stability of Sb₁Cu-5 catalysts, we used an H-cell as the reaction reactor, which could avoid flooding and carbonation problems in MEA. The FE_{CO} of approximately 90% during 100 h of continuous electrolysis and the stable cathode potential confirmed the impressive stability of the catalyst.

Supplementary Fig. 18 | Ex situ Sb K-edge EXAFS spectra of the Sb_1Cu_5 catalyst after 100 h of continuous electrolysis without phase correction. Sb foil is used as a reference. Peaks assigned to the Sb-Cu bond, together with the absence of the Sb-Sb bond, proved robust atomic Sb-Cu interfaces in the Sb_1Cu_5 catalyst after 100 hours of continuous electrolysis. In addition, the as-prepared sample and the sample after 30 min of electrolysis in the flow cell are also shown for comparison. The Sb_1Cu_5 catalyst after 100 hours of electrolysis in the H-cell showed almost the same Sb coordination structure as the as-prepared and post-catalysis samples, demonstrating the intrinsic long-term stability of the catalyst.

Comment 3

Expect for the activity, the selectivity of Cu was also modulated by the introduction of Sb. Thus, the discussion about catalytic performance should be modified.

Response

We appreciate the reviewer for his/her good suggestion. In the CO₂RR performance section, we updated our description to “This leads us to believe that the intrinsic activity and selectivity of Cu was significantly modulated by the introduction of isolated Sb-Cu atomic interfaces.”, to further highlight our modification in both activity and selectivity. In addition, we have strengthened our modification in selectivity compared with pure Cu in the abstract and conclusions sections. In the abstract section, we mentioned “..... the C-C coupling process on Cu significantly depletes CO* intermediates, producing many hydrocarbon

and oxygenate mixtures.” and “Herein, we report an antimony-copper single-atom alloy catalyst (Sb_1Cu) ca. 91% CO Faradaic efficiency, and negligible C_{2+} products are observed.” In the conclusion section, we said “Distinct from Cu with major C_{2+} products at high current density, high activity and selectivity towards CO production on Sb_1Cu-5 was elaborated”.

Comment 4

Additionally, catalytic performance of the Cu-Bi random alloy catalysts are very essential, which would be much helpful to reveal or confirm the role of isolated Sb-Cu atomic interface.

Response

We appreciate the reviewer for raising this important comment. We speculate that the reviewer refers to Cu-Sb random alloy. Accordingly, we synthesized a Cu-Sb random alloy using an electrodeposition method. Cu-Sb alloys were electrodeposited on carbon paper at a potential of -0.85 V vs. RHE for 5 min from 0.4 M citric acid aqueous solution containing 0.05 M $SbCl_3$ and 0.1 M $Cu(NO_3)_2$. The pH was adjusted to 6 by 5 M KOH. The atomic ratio of Cu:Sb was ca. 4:1, as determined by ICP-AES. STEM-EDS mapping showed even distribution of Cu and Sb, confirming the successful synthesis of the Cu-Sb random alloy (**Fig. R2**). In the CO_2RR performance test, the Cu-Sb random alloy showed $FE_{CO} < 50\%$ (e.g., ca. 45% at -100 mA cm^{-2}) (**Fig. R3**), much lower than that in Sb_1Cu-5 (ca. 95% of FE_{CO} at -100 mA cm^{-2}). Certain amount of hydrogen and formate were also found in the CO_2RR products on the Cu-Sb random alloy. Considering the few or even lack of isolated Sb-Cu interfaces in the Cu-Sb random alloy, the better CO_2RR performance on the Sb_1Cu-5 SAA than the Cu-Sb random alloy was presumably ascribe to the isolated Sb-Cu atomic interfaces. Therefore, this result further verified the importance of the isolated distributed Sb-Cu atomic interface.

Fig. R2 | STEM-EDS mapping of Cu-Sb random alloy. The image showed even distribution of Cu and Sb, confirming the formation of a Cu-Sb random alloy.

Fig. R3 | CO₂RR performance of Cu-Sb random alloy in a flow cell.

Comment 5

Since the efficient three-phase contact of CO₂ (gas), H₂O (liquid) and catalyst (solid) is of great importance for the performance, what about mass transfer?

Response

We thank the reviewer for raising this question. Mass transfer is an important factor in CO₂RR. To explore the factor of mass transfer in CO₂RR performance, we conducted the CO₂RR test in an H-cell and a flow cell with different mass transfer efficiency. In an H-cell, CO₂ gas is purged into the aqueous catholyte, and the dissolved CO₂ molecules are adsorbed on the electrocatalyst surfaces and undergo reduction (**Fig. R4a**). Achieving high rates of reaction in these conditions is limited given the low concentration (33 mM) and slow diffusion of aqueous CO₂ (diffusion coefficient $t_{\text{CO}_2} = 0.00176 \text{ mm}^2 \text{ s}^{-1}$ at 20°C) [*Nat. Energy* **7**, 130-143 (2022)]. In the H-cell, the FE_{CO} on the Sb₁Cu-5 catalyst maintained >80% at low current densities, and dramatically decreased at more negative potential due to the mass transport limitation that favored HER. In contrast, in a flow cell, the CO₂ gas diffuses through the back of the GDE-based cathode (**Fig. R4b**), where gaseous CO₂ is fed directly to an interface between the catalyst and electrolyte [*Chem. Soc. Rev.* **50**, 12897-12914 (2021)]. This facilitates the rapid mass transport of CO₂ to the catalyst surface, where it is bound and subjected to the proton and electron transfers necessary to form a given product. By taking advantage of flow cell, remarkably high CO partial current density was achieved for the Sb₁Cu-5 catalyst while maintaining a high CO selectivity above 90% (**Fig. R5**). The notably enhanced performance in the flow cell compared with H-cell is ascribed to its promoted CO₂ mass transfer [*Nat. Commun.* **12**, 1449 (2021); *Nat. Commun.* **13**, 6082 (2022); *Science* **364**, 1091-1094 (2019)]. In addition, some articles reported that increasing CO₂ mass transport by introducing ionomers [*Science* **367**, 661-666 (2020)] or designing hierarchical nanostructures [*Nat. Commun.* **13**, 3080 (2022)] could also enhance current density significantly. In summary, the improvement in mass transport is an important factor in the case of the Sb₁Cu-5 SAA catalyst for the CO₂RR, which results in significant enhancement in the CO₂RR activity.

Fig. R4 | Schematics of two CO₂RR reaction cells: a) traditional H-cell and b) flow cell. [Adv. Mater. 2103900 (2021)]

Fig. R5 | J-V curve of the CO partial current density against applied potential on the Sb₁Cu-5 catalyst in a flow cell and an H-cell.

Comment 6

It is suggested to give a radar chart when comparing with the previously reported state-of-the-art CO-selective electrocatalysts, which would be much easier to show the advantage.

Response

We appreciate the reviewer for his/her valuable suggestion. We made a radar chart to compare our catalyst with previously reported state-of-the-art CO-selectivity catalysts and

put it into the supplementary information (**Supplementary Fig. 23**, copied below). Sb₁Cu-5 catalyst exhibited a much higher CO current density, CO Faradaic efficiency and CO₂ conversion rate than the previously reported CO-selective catalysts.

Supplementary Fig. 23 | Comparison of the CO partial current density (j_{CO}), CO Faradaic efficiency (FE_{CO}) and conversion rate of CO₂-to-CO (Con.) with those of state-of-the-art CO-selective electrocatalysts. [Ref. R1: *Angew. Chem. Int. Ed.* **61, e202111683 (2022)]; [Ref. R2: *Nat. Commun.* **12**, 1449 (2021)]; [Ref. R3: *Angew. Chem. Int. Ed.* **60**, 11959-11965 (2021)]; [Ref. R4: *ACS Energy Lett.* **3**, 2835-2840 (2018)]; [Ref. R5: *Science* **365**, 367–369 (2019)]**

Comment 7

The experimental section needs to be far more detailed. It's far too limited for the work to be reproduced effectively. Please can the authors also include a chemicals (source and purity) section.

Response

We have updated the experimental section with more details and included the source and purity of the chemicals (copied below).

“Chemicals: Copper (II) chloride dihydrate (CuCl₂·2H₂O, ACS), antimony trichloride (SbCl₃, 99.9%), citric acid (H₃Cit, 99.5%), sodium borohydride (NaBH₄, 97%), hydrochloric

acid (HCl, GR), antimony powder (Sb, 99%) and isopropanol (IPA, 99.5%) were purchased from Macklin. All chemicals were used without further purification.

Synthesis: Sb_1Cu catalysts were synthesized by co-reduction of $CuCl_2$ and $SbCl_3$ with $NaBH_4$. $SbCl_3$ mixture solution was first prepared, including 1 mmol $SbCl_3$, 3 mmol H_3Cit and 20 ml deionized (DI) water. In the typical synthesis of the Sb_1Cu-5 catalyst, 3.2 ml $SbCl_3$ mixture solution, 2 ml 1 M $CuCl_2 \cdot 2H_2O$ solution, and 1 ml 3 M HCl were added into 13.8 ml DI water to obtain the precursor solution. The above solution was then rapidly added into 10 ml 1 M $NaBH_4$ solution in an ice bath and aged for 1 h. After violent reaction, the obtained black precipitate was then washed with DI water and IPA several times and dried under vacuum overnight. The obtained black powder was kept in a glove box under an Ar atmosphere for storage. For the synthesis of the $Sb_1Cu-1.5$ catalyst, 1.2 ml $SbCl_3$ mixture solution, 2 ml 1 M $CuCl_2 \cdot 2H_2O$ solution, and 1 ml 3 M HCl were added into 15.8 ml DI water to obtain the precursor solution. The following steps were the same as Sb_1Cu-5 .

For the synthesis of the Cu catalyst, the precursor solution was obtained by adding 2 ml 1 M $CuCl_2 \cdot 2H_2O$ solution and 1 ml 3 M HCl into 17 ml DI water. For the synthesis of the Sb_1Cu-10 catalyst, the precursor solution was obtained by adding 8 ml $SbCl_3$ mixture solution, 2 ml 1 M $CuCl_2 \cdot 2H_2O$ solution and 1 ml 3 M HCl into 9 ml DI water. The following steps were the same as the synthesis of Sb_1Cu .

Comment 8

Concerning the CO_2 -TPD results, whether the desorption peak of could be directly attributed to CO_2 ? Since the pretreatment temperature is only up to 400 °C, the peak around 400-500 °C needs to be determined in combination with MS results.

Response

We appreciate the reviewer for this very important suggestion. We repeated the CO_2 -TPD tests and added an MS sensor to verify that the peak at approximately 400°C was attributed to CO_2 desorption (**Fig. R6**). The MS signal showed the same peak at approximately 400°C, which matched well with the TCD results, confirming that the peak around 400°C was attributed to CO_2 desorption. However, we apologize for ignoring the possible reconstruction of Sb_1Cu-5 under high temperature. The XRD pattern showed peaks of Sb_2O_3 in Sb_1Cu-5 after treatment at 400°C, demonstrating phase separation of Sb and Cu metal in the Cu-Sb alloy (**Fig. R7**). Therefore, we deleted the TPD results in the manuscript.

Fig. R6 / MS signal ($m/z=44$) in CO_2 -TPD measurements of Cu and Sb_1Cu-5 samples.

Fig. R7 / XRD pattern of the Sb_1Cu-5 sample after treatment at 400°C.

Comment 9

It is suggested to add the results of CO desorption experiment.

Response

We appreciate the reviewer for this valuable suggestion. Considering the reconstruction of Sb_1Cu -5 under high temperature, we alternatively conducted *in situ* attenuated total reflection surface-enhanced infrared absorption spectroscopy (ATR-SEIRAS) measurements to compare the ability of CO desorption on Cu and Sb_1Cu samples at room temperature to avoid reconstruction during TPD tests. After obtaining $^*\text{CO}$ adsorbates, we suspended the applied potential and used Ar to purge into the electrolyte to sweep away $^*\text{CO}$ adsorbates (**Supplementary Fig. 26**, copied below). The attenuation rate and retention time of the ATR-SEIRAS signal of $^*\text{CO}$ could reflect the ability of CO desorption [*Science* **350**, 185-189 (2015)]. The attenuation rate was faster on Sb_1Cu -5 than on Cu, manifesting its lower binding energy of $^*\text{CO}$ and better ability of CO desorption, which matched well with spectroscopy measurements and theoretical calculation results.

Supplementary Fig. 26 | In situ ATR-SEIRAS spectra of a) Cu and b) Sb_1Cu -5 under an Ar sweep after suspension of the applied potential and c) attenuation of the $^*\text{CO}$ peak area with time. To further confirm the better ability of CO desorption on Sb_1Cu -5, we investigated the $^*\text{CO}$ retention time under an Ar sweep. The faster attenuation rate and shorter retention

time of *CO on Sb₁Cu-5 than Cu manifested its lower binding energy of *CO and better ability of CO desorption.

Comment 10

The authors stated that “...the formation of copper oxides was attributed to oxygen susceptibility of the Cu nanocrystal surface when exposed to air”. Then, how to guarantee that samples prepared from different batches are consistent. Whether this factor affect the catalytic performance?

Response

We appreciate the reviewer for raising this point. Generally, freshly synthesized Cu nanocrystals were susceptible to oxidation. To prevent further oxidation of the catalysts, after drying the powders in vacuum overnight, we placed the samples into a glove box under an Ar atmosphere for storage. In addition, under a CO₂RR environment, the catalysts were reduced to the metal state, without observation of oxides (**Supplementary Fig. 14**, copied below). Furthermore, the CO₂RR performance of the catalysts was repeatable, as shown in the error bar of FEs in **Fig. 2a** from different batches of samples (< ±5%). Therefore, we believe that under the protection of Ar atmosphere, further oxidation of as-prepared Sb₁Cu-5 was stifled which assures the catalytic performances on different batches of samples to be consistent.

Supplementary Fig. 14 | In situ Cu K-edge EXAFS spectra of the Sb₁Cu-5 catalyst under

CO₂RR conditions. Spectra of Sb₁Cu-5 under OCP (dashed line) and -1.0 V vs. RHE (solid line) are shown in the figure, with Cu, Cu₂O and CuO as references. The strengthened Cu-Cu peak (at approximately 2.50 Å) and disappearing Cu-O peak reconfirmed that metallic Cu was formed under the CO₂RR.

Reviewer 2

The authors study a copper electrocatalyst doped with slight amounts of Sb for the electrochemical reduction of CO₂ to CO. There are already many reports out on Cu based multimetallic systems for CO₂ electroreduction. Therefore, the approach the authors have taken is not extremely novel. The results are however interesting, as they show that CO can be produced selectively over a range of conditions, including high current density measurements in a MEA setup. The study mentions 1.5 and 5 % Sb in Cu, but does not mention (or investigate) what the optimal composition would be. Apparently there is a lower limit of Sb for the system to be selective to CO. Is there a upper limit for Sb concentration? Overall, the authors perform a very thorough investigation, with detailed material characterization, spectroelectrochemistry and DFT modeling. In general, I think this study can be considered for publication in Nature Communications.

Response

We highly appreciate the reviewer's high evaluation of our study.

As to the reviewer's concern that "Apparently there is a lower limit of Sb for the system to be selective to CO. Is there a upper limit for Sb concentration?", we demonstrated that there is an upper limit of Sb concentration for the Sb-Cu system to realize high selectivity of CO. In new experiments, we synthesized a Cu-Sb alloy using the co-reduction method with a higher content of Sb in the precursor solution. The concentration of Sb was *ca.* 10 at% (denoted as Sb₁Cu-10), as determined by ICP-AES. The HAADF-STEM image showed the formation of Sb aggregates in Sb₁Cu-10 (**Supplementary Fig. 13**, copied below) rather than even distribution of isolated Sb atoms in Sb₁Cu-5. As expected, due to the formation of Sb-Cu interfaces, negligible C₂₊ formation and >80% FE_{CO} were found on Sb₁Cu-10 (**Supplementary Fig. 24**, copied below), which confirmed the role of Sb-Cu interfaces in facilitating CO desorption and limiting C-C coupling. However, more formate produced on Sb₁Cu-10 compared with Sb₁Cu-5 was attributed to the formation of Sb clusters, considering that pure Sb exhibited relatively higher selectivity towards formate. This result manifested the importance of isolated Sb-Cu interfaces. In summary, to obtain high activity and selectivity towards CO₂-to-CO, an isolated Sb₁Cu interface was crucial. Neither too high (forming Sb aggregation and leading to formate production) nor too low (only modulating a

small portion of surface Cu and leading to C_{2+} production) Sb concentrations were demanded in the Cu-Sb system for selective CO_2 -to-CO conversion at a commercially relevant scale.

Supplementary Fig. 13 | HAADF-STEM image of the Sb_1Cu-10 catalyst. The black circle highlights Sb aggregation, confirming the formation of Sb clusters in the Sb_1Cu-10 catalyst.

Supplementary Fig. 24 | CO_2RR performance of the Sb_1Cu-10 catalyst in a flow cell. As expected, due to the formation of Sb-Cu interfaces, negligible C_{2+} formation and over 80% FE_{CO} were found on Sb_1Cu-10 , which confirmed the role of Sb-Cu interfaces in facilitating CO desorption and limiting C-C coupling. However, more formate produced on Sb_1Cu-10 compared with Sb_1Cu-5 was attributed to the formation of Sb clusters, considering that pure Sb exhibited relatively higher selectivity towards formate. This result manifested the importance of isolated Sb-Cu interfaces.

Reviewer 3

In this manuscript, the authors report that the Sb-Cu single-atom alloy catalyst constructed by a simple and mild method shows a high CO selectivity for the CO₂ reduction (CO₂RR) with a FE over 95%. Furthermore, a current density of 452 mA/cm² is realized in a designed flow cell. By combining online mass spectrometry, in-situ vibration spectroscopy and theoretical calculation, the high CO selectivity on Sb-Cu catalyst was also explored and explained. The authors propose that the isolated Sb-Cu interface is mainly accountable for the high CO selectivity. However, there have been a number of reports regarding Cu-Sb catalyzed electroreduction of CO₂ to CO with high selectivities despite that the authors ignore others' work for unknown reason. Considering these, the novelty of this paper is incremental but not substantial, therefore, in my opinion, this paper needs improvement before publication. Moreover, there are some other issues that should be addressed.

Response

We appreciate the reviewer's constructive comments. In this revised version of the manuscript, we have addressed all questions raised by the reviewer accordingly.

Here, we would like to emphasize the significance of this work, which distinguishes it from other works of Cu-Sb catalysts. In our work, we described isolated atomic Sb-Cu interfaces that modulated the Cu electronic structure and facilitated CO production, which was experimentally and theoretically corroborated. In contrast, other reported Cu-Sb bulk alloys, such as Cu₂Sb [*Nano Res.* **14**, 2831-2836 (2021)] and Sb_{0.22}Cu [*Appl. Catal. B: Environ.* **306**, 121089 (2022)], Cu and Sb formed a bulk alloy, which was used to investigate the bimetallic effect of Cu and Sb. However, these catalysts exhibited relatively low activity towards the CO₂RR. In addition, Sb-modified Cu electrodes synthesized through the galvanic replacement method [*ACS Catal.* **11**, 6846-6856 (2021)] incorporated Sb clusters on the Cu surface and investigated the modification effect of Sb clusters. Because of the formation of Sb aggregates, *ca.* a 10% FE of hydrogen and formate was found on the electrode, resulting in an FE_{CO} below 80%. In contrast, our work for the first time strengthened the effect of Sb single atoms in modulating the electronic structure of the Cu matrix and demonstrated the crucial role of atomic Sb-Cu interfaces experimentally and theoretically.

In addition, the comparison of the CO₂RR performance of Sb₁Cu-5 with recently reported Cu-Sb catalysts is shown in **Supplementary Table 4** (copied below). Sb₁Cu-5 achieved a CO partial current density above 450 mA cm⁻² with *ca.* 90% FE_{CO}, manifesting the excellent activity and selectivity towards CO₂-to-CO on Sb₁Cu interface sites. However, all the recently reported Cu-Sb catalysts failed to achieve CO partial current densities higher than 50 mA cm⁻², far from the commercially relevant scale. Therefore, the Sb₁Cu-5 catalyst exhibited the best CO₂RR performance among the reported Sb-Cu electrocatalysts.

Supplementary Table 4 | Performance of recently reported Sb-Cu electrocatalysts.

Catalyst	Potential (V vs. RHE)	j _{CO} (mA cm ⁻²)	FE _{CO} (%)	Ref.
Sb ₁ Cu-5	-1.16	452	90.4	This work
	-1.13	360	90.0	
Cu ₂ Sb NA/CF	-0.9	6	86.5	Nano Res. 14 , 2831-2836 (2021)
Sb _{0.22} Cu	-1.2	41	90	Appl. Catal. B: Environ. 306 , 121089 (2022)
Sb-Cu	-1.1	4.7	80	ACS Catal. 11 , 6846-6856 (2021)

Comment 1

How to define single-atom alloy catalysts? Traditionally, single-atom and alloy are two concepts. In Fig. 1, how to distinguish Sb single-atom from surface Sb-Cu single-atom alloy?

Response

Single-atom alloy (SAA) catalysts are a class of catalysts in which “one metal is atomically dispersed throughout the catalyst *via alloy bonding*” [*Chem. Soc. Rev.* **50**, 569-588 (2021); *ACS Catal.* **7**, 1491–1500 (2017)]. In the bulk alloys, two or more metals are combined together, with the formation of aggregates in each metal (**Fig. R8a**) [*Nano Today*, **34**, 100917 (2020)]. Unlike traditional bulk alloys, SAA involves the use of a low concentration of dopant atoms, which exist in single-atom form in the host metal (**Fig. R8b**) [*Chem. Rev.* **120**, 12044–12088 (2020); *Nat. Commun.* **13**, 3188 (2022); *Nat. Chem.* **10**, 325-332 (2018)]. According to the definition, two main criteria are crucial in identifying SAAs: the dopant atoms are atomically dispersed in the host metal; metallic bonds formed between the dopant and the host metals.

To distinguish Sb single-atom from surface Sb-Cu single-atom alloy in the catalyst, we used high-angle annular dark-field scanning transmission electron microscopy (HAADF-STEM) and extended X-ray absorption fine structure (EXAFS) as evidences as shown in **Fig. 1**. HAADF-STEM is a critical technique to identify single atoms in SAA. HAADF-STEM is based on Rutherford scattering, in which the image intensity for given atoms is roughly proportional to the square of the atomic number (Z^2) of the element, allowing heavy metal atoms to brightly contrast against low background supports. Since Sb is a heavier atom than Cu, we believe that all the bright spots represent isolated Sb atoms. HAADF-STEM clearly identified isolated bright spots representative of atomically dispersed Sb atoms in the Cu matrix for $\text{Sb}_1\text{Cu}-5$ (**Fig. 1b**, copied below). In addition, the formation of Sb-Cu bond was also verified by Sb K -edge EXAFS results (**Fig. 1d**, copied below), confirming Sb and Cu are combined by alloy bonding. Considering above evidences, $\text{Sb}_1\text{Cu}-5$ belonged to Sb-Cu single-atom alloy.

Fig. R8 / Schematic images of a) bulk alloy and b) single-atom alloy (SAA).

Enlarged Fig. 1b | HAADF-STEM image of the as-prepared Sb₁Cu-5 catalyst. The black circles highlight single Sb atoms.

Enlarged Fig. 1d | Ex situ EXAFS at the Sb K-edge of the as-prepared and post-catalysis Sb₁Cu-5 without phase correction. Sb foil and Sb₂O₃ are shown as references. The observation of Sb-Cu bond and the absence of Sb-Sb bond confirmed the formation of isolated atomic Sb-Cu interfaces.

Comment 2

At present, the optimal Sb loading mentioned in the article is about 5%. Is it possible for the Sb loading to be increased significantly so that more Sb-Cu interfaces can be constructed in favor of CO₂ to CO conversion?

Response

We appreciate the reviewer for raising this valuable point. To investigate the result of high Sb loading, we synthesized a Cu-Sb alloy using the co-reduction method with a higher content of Sb in the precursor solution. The concentration of Sb was *ca.* 10 at% (denoted as Sb₁Cu-10), as determined by ICP-AES. The HAADF-STEM image showed Sb aggregation in Sb₁Cu-10 (**Supplementary Fig. 13**, copied below) rather than even distribution of isolated Sb atoms in Sb₁Cu-5. As expected, due to the formation of Sb-Cu interfaces, negligible C₂₊ formation and >80% FE_{CO} were found on Sb₁Cu-10 (**Supplementary Fig. 24**, copied below), which confirmed the role of Sb-Cu interfaces in facilitating CO desorption and limiting C-C coupling. However, more formate produced on Sb₁Cu-10 compared with Sb₁Cu-5 was attributed to the formation of Sb clusters, considering that pure Sb exhibited

relatively higher selectivity towards formate. This result manifested the importance of isolated Sb-Cu interfaces. In summary, to obtain high activity and selectivity towards CO₂-to-CO, an isolated Sb₁Cu interface was crucial. Neither too high (forming Sb aggregation and leading to formate production) nor too low (only modulating a small portion of surface Cu and leading to C₂₊ production) Sb concentrations were demanded in the Cu-Sb system for selective CO₂-to-CO conversion at a commercially relevant scale.

Supplementary Fig. 13 | HAADF-STEM image of the Sb₁Cu-10 catalyst. The black circle highlights Sb aggregation, confirming the formation of Sb clusters in the Sb₁Cu-10 catalyst.

Supplementary Fig. 24 | CO₂RR performance of the Sb₁Cu-10 catalyst in a flow cell. As expected, due to the formation of Sb-Cu interfaces, negligible C₂₊ formation and over 80% FE_{CO} were found on Sb₁Cu-10, which confirmed the role of Sb-Cu interfaces in facilitating CO desorption and limiting C-C coupling. However, more formate produced on Sb₁Cu-10 compared with Sb₁Cu-5 was attributed to the formation of Sb clusters, considering that pure Sb exhibited relatively higher selectivity towards formate. This result manifested the importance of isolated Sb-Cu interfaces.

Comment 3

The authors are suggested to run the CO-TPR experiment in addition to the CO₂-TPR.

Response

We appreciate the reviewer for this important suggestion. We apologize for ignoring the possible reconstruction of Sb₁Cu-5 under high temperature. The XRD pattern showed peaks of Sb₂O₃ in Sb₁Cu-5 after treatment at 400°C, demonstrating phase separation of Sb and Cu metal in the Cu-Sb alloy (**Fig. R7**). Therefore, we deleted the TPD results in the manuscript. We further conducted *in situ* ATR-SEIRAS measurements to compare the ability of CO desorption on Cu and Sb₁Cu samples at room temperature to avoid reconstruction during TPD tests. After obtaining *CO adsorbates, we suspended the applied potential and used Ar to purge into the electrolyte to sweep away *CO adsorbates (**Supplementary Fig. 26**, copied below). The attenuation rate and retention time of the ATR-SEIRAS signal of *CO could reflect the ability of CO desorption [*Science* **350**, 185-189 (2015)]. The attenuation rate was faster on Sb₁Cu-5 than on Cu, manifesting its lower binding energy of *CO and better ability of CO desorption, which matched well with spectroscopy measurements and theoretical calculation results.

Fig. R7 | XRD pattern of the Sb_1Cu-5 sample after treatment at $400^\circ C$.

*Supplementary Fig. 26 | In situ ATR-SEIRAS spectra of a) Cu and b) Sb₁Cu-5 under an Ar sweep after suspension of the applied potential and c) attenuation of the *CO peak area with time. To further confirm the better ability of CO desorption on Sb₁Cu-5, we investigated the *CO retention time under an Ar sweep. The faster attenuation rate and shorter retention time of *CO on Sb₁Cu-5 than Cu manifested its lower binding energy of *CO and better ability of CO desorption.*

Comment 4

As Fig. 4f, Supplementary Fig. 22, and spectroscopy experimental, the CO is adsorption the top site of Cu site. And TS geometrical structure of C-C coupling is deviated the site, obviously. The authors are suggested to inset the IS and FS structures in Fig. 4d not only TS. And the reaction activity of different Cu site which close or far Sb atom can further discuss to explain the adsorption site of CO and other intermediates.

Response

We appreciate the reviewer for this suggestion. *In situ* spectroscopic measurements are not contradictory to theoretical calculations. It was found by DFT calculations that the most stable adsorption for a single CO* adsorption is also atop-CO* on both Cu (211) and Sb₁Cu-5 (211). However, the coadsorption of two CO* must be different from that of two single CO*-atops for CO-CO coupling (consistent with the *J. Phys. Chem. Lett.* **10**, 533-539 (2019)). The deviated CO* adsorption is only present for CO-CO coupling, which is close to a transient state. In other words, the lifetime is relatively short. Hence, the dominant presence of CO* adsorption is still atop the site. This is not contradictory to the experimental spectra.

The IS and FS structures of CO*-CO* coupling have been added and are now shown in **Fig. 4d** (copied below).

Enlarged Fig. 4d | Barriers for CO-CO* coupling on Cu (211) and Sb₁Cu-5 (211). The initial (IS), transition (TS) and final (FS) structures are shown as insets, where Cu, Sb, C, O and H are represented in orange, purple, gray, red, and white, respectively.*

The Cu site nearest to the Sb atom has been discussed in our original manuscript. The Cu sites far from the Sb atom should be very similar to pure Cu. We further studied the reaction activity of the Cu site, which is the next-nearest to the Sb atom on Sb₁Cu-5 (211), as shown in **Supplementary Fig. 30** (copied below). As COOH* and HCOO* prefer to be adsorbed at the bridge site of Cu, the adsorption energies were close for the Cu sites, either nearest or next-nearest to the Sb atom. However, the adsorption energy of CO* at the top site on Cu (next-nearest to Sb atom, -0.21 eV) was close to pure Cu (-0.20 eV), more stable than CO* at the top site of Cu (nearest to Sb atom, -0.11 eV).

On the Cu site (next-nearest to the Sb atom), CO is also the main product due to the lower barrier compared to HCOOH and C₂₊ formation. However, CO* accounts for 77% of the total sites for CO₂RR. This is inconsistent with the results of spectroscopic measurements, where the lower frequency of the CO* peak on Sb₁Cu-5 implied weakened CO* adsorption and much lower CO coverage relative to Cu (**Fig. 3d**). Hence, the Cu sites (next-nearest to Sb atom) on Sb₁Cu-5 should not be the main active sites. The Cu sites (nearest to Sb atom) on Sb₁Cu-5 should have the major activity contributions.

Supplementary Fig. 30 | CO₂RR to CO and HCOOH on Cu site (next-nearest to Sb atom) on Sb₁Cu-5 (211). The adsorption structures are shown on the right, where Cu, Sb, C, O and H are represented in orange, purple, gray, red and white, respectively.

Comment 5

The detail reaction free energies are calculated, while, in Fig. 4f, the authors need explain the reason of the greatly geometrical structures different between FS₁ and IS₂, it not accomplishes of one structure for COOH intermediate at this reaction?*

Response

We appreciate the reviewer for raising this valuable point. The structures of COOH* in CO₂ protonation and CO* formation are different because the water layers of these two states were different. For FS₁, the proton from the water layer has been added to COOH*. It is a neutral water structure. However, for IS₂, the water structure is before protonation. In other words, it is a charged water structure. The differences in water layers between FS₁ and IS₂ lead to the structural transformation of COOH*.

Comment 6

The obvious Stark slope of CO is obvious, why not the adsorption energy of CO is not influenced under different potentials at Fig. 4f.*

Response

In **Fig. 4f**, a computational hydrogen electrode model was used to calculate the free energy change at varying potentials. The major free energy differences at varying potentials are from the electrochemical process, namely, the chemical potential variation of the electron-proton pair. The free energy changes of CO₂ protonation and COOH* protonation (CO* formation) can be significantly affected by potentials, where proton transfer occurs, as shown in **Fig. 4f**. However, reactions without proton transfer, such as CO adsorption or desorption, are usually considered potentially independent [*Nat. Nanotechnol.* **16**, 1386-1393 (2021)]. Hence, the adsorption energy of CO* is not significantly influenced under different potentials.

We have also strictly investigated the potential effects on the adsorption energy of CO*. It also supported our explanations above. An electric field was applied in DFT calculations. Based on a parallel-plate capacitor model, a linear correlation between the electric field and absolute potential was approximated as follows [*J. Phys. Chem. C* **124**, 14581-14591 (2020)]:

$$E = \frac{\sigma}{\varepsilon\varepsilon_0} = \frac{C_H(U_{SHE} - U_{PZC})}{\varepsilon\varepsilon_0}$$

where σ is the surface charge density and ε and ε_0 are the dielectric constants of vacuum and water near the interface, which were set to $8.85 \times 10^{-12} \text{ F m}^{-1}$ and 2 (unitless), respectively. C_H refers to the Helmholtz capacitance ($\mu\text{F cm}^{-2}$), which was set to $25 \mu\text{F cm}^{-2}$. U_{SHE} is the electrode potential referenced to a standard hydrogen electrode (SHE). U_{PZC} refers to the potential of zero charge (PZC) versus SHE, which was set to -0.9 V [*J. Phys. Chem. Lett.* **11**, 9802-9811 (2020)].

The SHE could be related to the RHE by the following formula:

$$U_{RHE} = U_{SHE} + 0.059\text{pH}$$

The calculated adsorption energies of CO* changed very little ($< 0.05 \text{ eV}$) with potentials between 0 and -1.2 V vs. RHE , which showed consistent trends for Cu (211) and Sb₁Cu-5 (211). All insights and conclusions shown in the original manuscript are still reliable. In addition, by varying potentials from -0.2 to -0.7 V vs. RHE , the calculated adsorption energies of CO* weakened from -0.13 to -0.11 eV on Sb₁Cu-5 (211) and from -0.23 to -0.20 eV on Cu (211), which showed consistent trends with the results of the *in situ* ATR-SEIRAS spectra (**Fig. 3b-d**).

Additional microkinetic modeling was performed to double check the reliability of the present kinetic analysis. All conclusions are not affected. Overall, the potential effects on the adsorption energy of CO* were ignored in the calculation of free energy change.

In response to the Reviewer's comment, we have added the electric field effect on the adsorption energy of CO* in the **Computational details**.

Comment 7

The authors are suggested to compare the relevant work in the literature with their own.

Response

We now have compared Sb₁Cu-5 with other recently reported Cu-based catalysts or non-Cu-based catalysts in **Figs. 2d, 2e, Supplementary Tables 2 and 3** (copied below). Sb₁Cu-5 achieved a CO partial current density above 450 mA cm⁻² with *ca.* 90% FE_{CO}, outperforming the previously reported state-of-the-art CO-selective electrocatalysts, manifesting the excellent activity and selectivity towards CO₂-to-CO on isolated Sb-Cu interface sites. We also compared our work with other Sb-Cu catalysts (**Supplementary Table 4**, copied below). All the recently reported Cu-Sb catalysts failed to achieve CO partial current densities higher than 50 mA cm⁻², far from the commercially relevant scale.

Enlarged Fig. 2d | Performance metrics of different reported CO₂RR-to-CO Cu-based catalysts.

Enlarged Fig. 2e | Performance metrics of different reported CO₂RR-to-CO non-Cu-based catalysts.

Supplementary Table 2 | Performance of recently reported non-Cu-based CO₂-to-CO electrocatalysts in flow cells.

Catalyst	Potential (V vs. RHE)	j_{CO} (mA cm ⁻²)	FE _{CO} (%)	Ref.
Sb ₁ Cu-5	-1.16	452	90.4	This work
	-1.13	360	90.0	
Ni-SA/PCFM	-1.2	337	81	Nat. Commun. 11 , 593 (2020)
Fe ³⁺ -N-C	-0.45	94	94	Science 164 , 1091-1094 (2019)
CoPc@Fe-N-C	-0.83	277	94	Adv. Mater. 31 , 1903470 (2019)
Ni@NiNCM	-0.92	126	84	Angew. Chem. Int. Ed. 60 , 11959-11965 (2021)
CoPc	/	172	86	Science 365 , 347-369 (2019)
Mg-C ₃ N ₄	-0.61	270	90	Angew. Chem. Int. Ed. 60 , 25241-25245 (2021)
Zn/NC-NSs	-1.06	67	84	Angew. Chem. Int. Ed. 61 , e202111683 (2022)

Supplementary Table 3 | Performance of recently reported Cu-based CO₂-to-CO electrocatalysts.

Catalyst	Potential (V vs. RHE)	j_{CO} (mA cm ⁻²)	FE _{CO} (%)	Ref.
----------	-----------------------	--	----------------------	------

Sb₁Cu-5	-1.16	452	90.4	This work
	-1.13	360	90.0	
Cu/Ni(OH) ₂	-0.5	3.7	92	Sci. Adv. 3 , 9 (2017)
Cu-APC	-0.78	8.6	92	Nat. Chem. 11 , 222-228 (2019)
Cu-S ₁ N ₃ /Cu _x	-0.75	7.5	90	Angew. Chem. Int. Ed. 60 , 24022–24027 (2021)
V-CuInSe ₂	-0.6	70	92	Adv. Mater. 34 , 2106354 (2022)
CuCo _{1.0}	/	60.5	97.4	ACS Sustain. Chem. Eng. 8 , 12561–12567 (2020)
CuZn NW	-1.0	14.4	90	ACS Catal. 12 , 2741–2748 (2022)
Cu ₉₇ Sn ₃	-0.45	120	87	Nat. Commun. 12 , 1449 (2021)
Cu/Cu ₂ O-Sb-5	/	110	91	J. Mater. Chem. A 9 , 23234 (2021)

Supplementary Table 4 | Performance of recently reported Sb-Cu electrocatalysts.

Catalyst	Potential (V vs. RHE)	j_{co} (mA cm⁻²)	FE_{co} (%)	Ref.
Sb₁Cu-5	-1.16	452	90.4	This work
	-1.13	360	90.0	
Cu ₂ Sb NA/CF	-0.9	6	86.5	Nano Res. 14 , 2831-2836 (2021)
Sb _{0.22} Cu	-1.2	41	90	Appl. Catal. B: Environ. 306 , 121089 (2022)
Sb-Cu	-1.1	4.7	80	ACS Catal. 11 , 6846-6856 (2021)

Comment 8

Some other minor points:

Check the typos throughout the manuscript, e.g., Line 107, NaHB₄.

The label of Figure S20 is messed up, a is Cu, b is Sb-Cu?

Check the format of references, especially journal abbreviations.

The Sb₁Cu-5, and the Sb₁Cu-1.5, What does subscript 1 mean?

Response

We appreciate the reviewer for pointing out these points. We are sorry that we made some mistakes in the manuscript and supplementary information. We have corrected “NaHB₄” to “NaBH₄” in line 107 and checked the typos throughout the manuscript. The labels in Supplementary Fig. 20 were messed up. Figure a shows the H-cell performance on the Sb₁Cu-5 catalyst, while Figure b shows the H-cell performance on the Cu catalyst. The corrected figure is copied below. In addition, we have changed the format of references to the standard format of *Nature Communications*. When we describe the SAA catalyst, we use A₁B to indicate that A atoms are isolated and dispersed in the B metal host (e.g., Pb₁Cu [*Nat. Nanotechnol.* **16**, 1386-1393 (2021)], Pd₁Ni [*Nat. Commun.* **10**, 4998 (2019)]). Here, the subscript “1” means that Sb atoms are isolated and dispersed in the Cu host, forming atomic Sb-Cu interfaces.

Supplementary Fig. 25 | a, b) H-cell performance and c) Tafel plot of Sb₁Cu-5 and Cu catalysts.

REVIEWERS' COMMENTS

Reviewer #1 (Remarks to the Author):

Thanks the authors for taking into consideration of all my comments. The modifications are sufficient and reasonable. I can now suggest it for publication on Nature Commination.

Reviewer #3 (Remarks to the Author):

I have checked the revised manuscript and the author's response. I think the authors have made large effort to response all the comments and questions. In the responses, the authors compares the catalyst performance made by themselves with the reported CuSb, highlighting the innovation of the article. Most reports on the performance of CuSb catalysts are carried out in H-type electrolyzer. The authors need to quote the relevant CuSb articles reasonably in the manuscript, and compare the performance of catalysts under the same experimental conditions. I think the revised manuscript should be proper for publication after the authors further improved it.

Reviewer 1

Thanks the authors for taking into consideration of all my comments. The modifications are sufficient and reasonable. I can now suggest it for publication on Nature Commination.

Response

We highly appreciate the reviewer for time and insightful comments on our work.

Reviewer 3

I have checked the revised manuscript and the author's response. I think the authors have made large effort to response all the comments and questions. In the responses, the authors compares the catalyst performance made by themselves with the reported CuSb, highlighting the innovation of the article. Most reports on the performance of CuSb catalysts are carried out in H-type electrolyzer. The authors need to quote the relevant CuSb articles reasonably in the manuscript, and compare the performance of catalysts under the same experimental conditions. I think the revised manuscript should be proper for publication after the authors further improved it.

Response

We appreciate the reviewer's valuable suggestions. We have cited the relevant Cu-Sb articles in the manuscript (ref. 37-39). We have also compared the CO₂RR performance of Sb₁Cu-5 with recently reported Cu-Sb catalysts in an H-cell (**Supplementary Table 4**, copied below). In our work, Sb₁Cu-5 achieved above 90% FE_{CO} at a current density of -10 mA cm⁻² with the stability of 100 hours in an H-cell. Both the CO partial current density and FE_{CO} of Sb₁Cu-5 are higher than those of reported Cu-Sb catalysts (*e.g.*, Cu₂Sb NA/CF and Sb modified Cu). Because of different mass loading in the reported Cu-Sb articles, we also compared their mass activity normalized by the catalyst mass. Sb₁Cu-5 exhibited the highest mass activity among the reported Cu-Sb catalysts. In addition, the durability of Sb₁Cu-5 was much longer than other Cu-Sb catalysts. Therefore, the Sb₁Cu-5 catalyst exhibited the best CO₂RR performance among the reported Sb-Cu electrocatalysts.

Supplementary Table 4 | Performance of recently reported Sb-Cu electrocatalysts in an H-cell.

Catalyst	Potential (V vs. RHE)	j_{CO} (mA cm⁻²)	Mass activity (mA mg⁻¹)	FE_{CO} (%)	Stability (h)	Ref.
Sb ₁ Cu-5	-0.85	9.2	9.2	92.3	100	This work
Cu ₂ Sb NA/CF	-0.9	6	-	86.5	2	Nano Res. 14 , 2831-2836 (2021)
Sb _{0.22} Cu	-0.8	16.2	5.4	95	10	Appl. Catal. B: Environ. 306 , 121089 (2022)
Sb-Cu	-1.1	4.7	2.9	80	12	ACS Catal. 11 , 6846-6856 (2021)